# PreFM: Online Audio-Visual Event Parsing via Predictive Future Modeling

**Xiao Yu**[1,2]   **Yan Fang**[1,2]   **Yao Zhao**[1,2]   **Yunchao Wei**[1,2,✉]

[1]Institute of Information Science, Beijing Jiaotong University
[2]Visual Intelligence + X International Joint Laboratory
[✉]Corresponding Author
xiaoyu@bjtu.edu.cn   wychao1987@gmail.com

## Abstract

Audio-visual event parsing plays a crucial role in understanding multimodal video content, but existing methods typically rely on offline processing of entire videos with huge model sizes, limiting their real-time applicability. We introduce Online Audio-Visual Event Parsing (On-AVEP), a novel paradigm for parsing audio, visual, and audio-visual events by sequentially analyzing incoming video streams. The On-AVEP task necessitates models with two key capabilities: (1) Accurate online inference, to effectively distinguish events with unclear and limited context in online settings, and (2) Real-time efficiency, to balance high performance with computational constraints. To cultivate these, we propose the **Pre**dictive **F**uture **M**odeling (PreFM) framework featured by (a) predictive multimodal future modeling to infer and integrate beneficial future audio-visual cues, thereby enhancing contextual understanding and (b) modality-agnostic robust representation along with focal temporal prioritization to improve precision and generalization. Extensive experiments on the UnAV-100 and LLP datasets show PreFM significantly outperforms state-of-the-art methods by a large margin with significantly fewer parameters, offering an insightful approach for real-time multimodal video understanding. Code is available at https://github.com/XiaoYu-1123/PreFM.

## 1   Introduction

Multimodal learning [5, 72, 38, 79] is a significant topic in the machine learning research area. Among various modalities, audio [62] and vision [60, 47] are the primary ways humans perceive the world, making audio-visual learning (AVL) [19, 42, 35, 15] essential. Among various progress [43, 45, 31, 34] related to AVL, audio-visual event parsing (AVEP), i.e., understanding events in videos, becomes increasingly important with the explosive growth of video content on streaming platforms.

AVEP involves processing both modality-aligned (audio-visual) and modality-misaligned (audio-only or visual-only) events in video content. Prevailing methods [13, 14, 78] operate offline, analyzing entire video sequences to utilize global context for accurate video events understanding. Though offering precise predictions, the necessity of whole-video processing, often coupled with large models and consequently high computational costs, makes these approaches unsuitable for real-time applications that require immediate detection and swift responses in dynamic environments such as autonomous driving [65, 70], wearable devices [26, 2], and human-robot interaction [50, 32].

To tackle these limitations, we introduce **On**line **A**udio-**V**isual **E**vent **P**arsing (On-AVEP), a new paradigm that parses audio, visual, and audio-visual events in streaming videos with an online processing manner. The core characteristic of On-AVEP is to perceive the environmental state and generate timely feedback using only historical and current multimodal information, while balancing model performance and efficiency particularly in resource-aware and dynamic environments.

39th Conference on Neural Information Processing Systems (NeurIPS 2025).

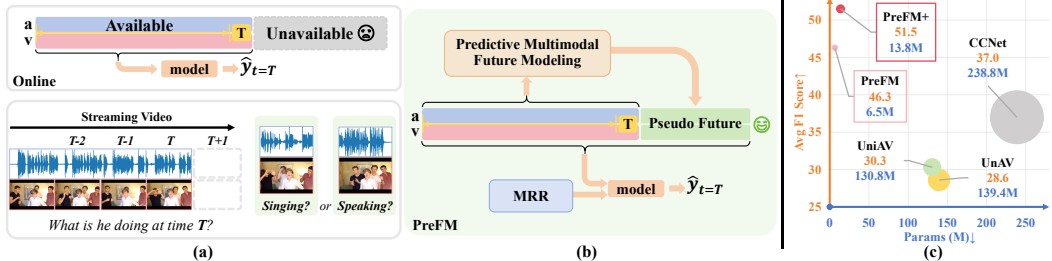

Figure 1: (a) Illustration of parsing events in online scenarios: if a man opens his mouth and produces a vocal sound at time $T$, it is unclear based solely on information from $0$ to $T$ whether this marks the beginning of a musical phrase (as part of "*singing*") or the start of a conversation (as in "*speaking*"). Precisely parsing events with these limited context is crucial for accurate online inference. (b) Simplified architecture of our PreFM framework, highlighting predictive future modeling and modality-agnostic robust representation (MRR). (c) Comparison of performance and efficiency against SOTA methods on the UnAV-100 [13] dataset.

Specifically, On-AVEP necessitates the model possess two key capabilities: (1) **Accurate Online Inference**: This requires that the model adapts to complex and dynamic scene variations and accurately predicts ongoing events by relying exclusively on past and current information without any future context. As illustrated in Figure 1(a), the model needs to distinguish similar events with unclear, limited context due to the lack of future information. (2) **Real-time Efficiency**: To meet the immediate response demands of On-AVEP applications, the model needs to achieve accurate event parsing with low computational cost, balancing performance and complexity well to satisfy the needs of online video processing.

To cultivate these essential capabilities, we introduce the **Pre**dictive **F**uture **M**odeling (PreFM) framework as illustrated in Figure 1(b). PreFM aims to predict future states through effective temporal-modality feature fusion and leverage knowledge distillation and temporal prioritization for training efficiency. To achieve (1) accurate online inference, PreFM employs **predictive multimodal future modeling**, using available data and fusing their features to infer beneficial future audio-visual cues. The cross-temporal and cross-modal feature interactions are utilized to effectively reduce noise within the pseudo-future context and enhance current representations. For (2) balancing real-time efficiency and overall parsing performance, PreFM integrates two designs during training: **modality-agnostic robust representation** distills rich, modality-agnostic knowledge from a large pre-trained teacher model for more generalized representation, and **focal temporal prioritization** encourages the model to focus on the most temporally critical information for online decisions, thereby boosting the model's inference accuracy while keeping high inference efficiency.

Extensive experiments on two challenging datasets, UnAV-100 [13] and LLP [54], demonstrate that the PreFM framework significantly outperforms existing state-of-the-art (SOTA) methods in both segment-level and event-level metrics. Moreover, PreFM exhibits substantial advantages in model efficiency, striking a superior balance between performance and model complexity, with the margin of $+\mathbf{9.3}$ in event-level average F1 score and merely $\mathbf{2.7\%}$ parameters as highlighted in Figure 1(c).

In summary, our main contributions are:

- **(I)** We introduce Online Audio-Visual Event Parsing (On-AVEP), a new paradigm for real-time multimodal understanding. To our knowledge, this is the first work to systematically address the challenge of parsing audio, visual, and audio-visual events from streaming video. We further establish that success in this paradigm requires two critical capabilities: (a) accurate online inference from limited context, and (b) real-time efficiency to balance performance with computational cost.

- **(II)** We propose the PreFM framework, a novel and efficient architecture for On-AVEP. PreFM's core innovations include: (a) Predictive Multimodal Future Modeling mechanism to overcome the critical problem of missing future context; and (b) a combination of Modality-agnostic Robust Representation and Focal Temporal Prioritization to enhance model robustness and efficiency during training, providing an insightful approach to multimodal real-time video understanding.

- **(III)** We establish new SOTA performance with unprecedented efficiency. Extensive experiments on two public datasets show that PreFM drastically outperforms previous methods (e.g., +9.3 Avg F1-score on UnAV-100), while using a fraction of the computational resources (e.g., only 2.7% of the parameters of the next best model), validating it as a powerful and practical solution.

## 2   Related Work

**Online Video Understanding**   encompasses online action detection for identifying actions [56, 7, 44], action anticipation for predicting future [6, 73], and online temporal action localization [48, 51] for determining action boundaries. Frameworks like JOADAA [17], TPT [67] and MAT [58] jointly model detection and anticipation tasks, bridging the present and future. Recent research focuses on model reliability through uncertainty quantification [18] and adaptability through open-vocabulary detection [61]. Concurrent advancements explore leveraging large language models for complex online understanding tasks [33, 3, 69]. However, these methods rely solely on the visual modality and neglect the crucial auditory perception, motivating our research into online audio-visual event parsing aiming to integrate both sensory streams for a more robust and holistic real-time understanding.

**Audio Visual Video Parsing (AVVP)**   aims to temporally classify videos within segments as audible or visible events. Early weakly-supervised methods [54, 66] use attention to infer temporal structure. Subsequent works [10–12, 4] further address modality imbalance and interaction. A significant recent trend involves leveraging external knowledge, using language prompts [9] or pre-trained models like CLIP [46]/CLAP [8] to denoise or generate finer pseudo-labels from weak supervision [29, 77, 30]. Building on this, methods such as CoLeaF [49], NREP [27], and MM-CSE [71] focus on sophisticated feature disentanglement and interaction for improved performance.

**Audio Visual Event Localization (AVEL)**   is first introduced to temporally locate events that are both visually and auditorily present within trimmed video clips [53]. Subsequent methods [63, 41, 28, 36, 23, 74] leverage cross-modal attention, background suppression, contrastive smaples and adapters to improve localization accuracy. AVE-PM [37] is developed to handle portrait-mode short videos, while OV-AVEL [75] extends the task into an open-vocabulary setting. For densely annotated, untrimmed videos featuring multiple overlapping events, UnAV [13] releases the UnAV-100 benchmark and inspires models like UniAV [14], LOCO [64], FASTEN [39] and CCNet [78], which employ multi-temporal fusion, local correspondence correction and cross-modal consistency for dense event localization. Recent efforts [68, 15, 52, 40] also aim to omni-understanding using powerful large language models. However, these approaches generally rely on full-video inputs and huge model sizes, making them unsuitable for real-time parsing. Our work distinguishes itself by unifying AVEL and AVVP into a comprehensive online audio-visual event parsing framework, designed for efficient real-time processing and capable of identifying events regardless of whether they are solely auditory, visual or audio-visual.

## 3   Methods

In this section, we first introduce the problem setup in Sec. 3.1 and present a brief overview of our method in Sec. 3.2, with core designs: Predictive Multimodal Future Modeling (Sec. 3.3), Modality-agnostic Robust Representation (Sec. 3.4), and Focal Temporal Prioritization (Sec. 3.5). Finally, we discuss the specifics of our approach during training and online inference in Sec. 3.6.

### 3.1   Preliminaries

On-AVEP involves predicting events within streaming videos by sequentially processing multimodal information. This task is primarily divided into two sub-tasks: online audio-visual event localization (On-AVEL) and online audio-visual video parsing (On-AVVP). In the former, given a sequence of audio-visual data pairs $\{V_t, A_t\}_{t=1}^{T}$ and the corresponding label $y_{t=T}$, where $T$ denotes the current time step, the model is required to predict the multi-label event vector $\hat{y}_{t=T} \in \{0, 1\}^{C_{av}}$, where $C_{av}$ represents the total number of audio-visual event categories. While the latter task involves predicting $\hat{y}_{t=T} \in \{0, 1\}^{C_a + C_v}$, where $C_a$ and $C_v$ represent the number of audio-only and visual-only events, respectively. In both sub-tasks, models typically take the pre-processed visual-audio feature vectors

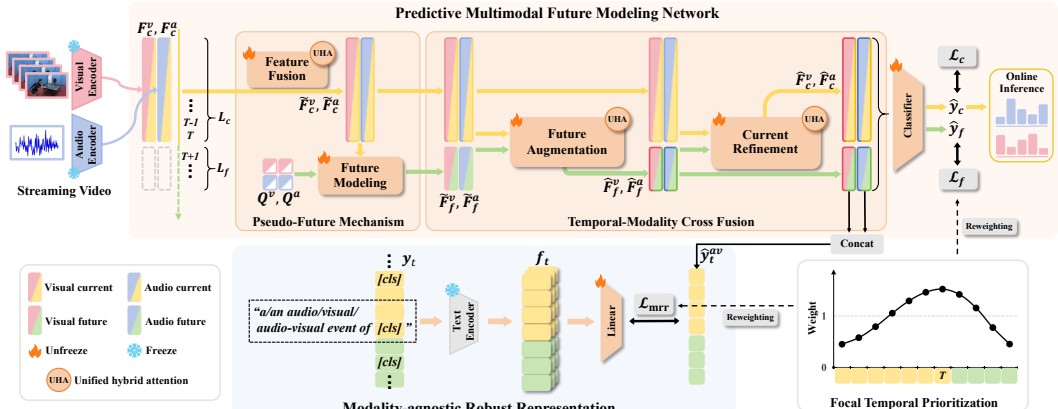

Figure 2: The pipeline of PreFM. It takes real-time audio-visual streams, using predictive modeling to generate multimodal future context, modality-agnostic robust representation to enhance performance by transferring knowledge, and focal temporal prioritization to emphasize the current time step $T$.

$\{f_t^a\}_{t=1}^T$ and $\{f_t^v\}_{t=1}^T \in \mathbb{R}^{T \times D}$ ($D$: feature dimension) within existing video datasets [54, 25, 53, 13] for subsequent operations.

## 3.2 Overview

As illustrated in Figure 2, during online inference, PreFM sequentially processes incoming audio and visual features $F_c^a, F_c^v$ available up to the current time $T$. To address the challenge of missing future information, which is crucial for event disambiguation, the core **Predictive Multimodal Future Modeling** network (Sec. 3.3) dynamically generates pseudo-future multimodal sequences. This process starts with a *Pseudo-Future Mechanism* that fuses current-time multimodal features and subsequently models initial pseudo-future predictions $\tilde{F}_f^a, \tilde{F}_f^v$, then *Temporal-Modality Cross Fusion* where pseudo-future cues and current representations are mutually enhanced through comprehensive cross-temporal and cross-modal interactions. The resulting contextually augmented representations $\hat{F}_c^a, \hat{F}_c^v$ are then utilized for event parsing at time $T$.

To train an effective and efficient PreFM model, in addition to direct supervision on predictions for both the current window and the pseudo-future sequences, PreFM utilizes **Modality-agnostic Robust Representation** (MRR, Sec. 3.4). Through MRR, event labels $y_t$ are transformed into target modality-agnostic features $f_t$ using a pre-trained teacher model; PreFM's internal event representations $\hat{y}_t^{av}$ are then guided to align with these target features via a dedicated distillation loss term. Furthermore, **Focal Temporal Prioritization** (Sec. 3.5) is implemented by reweighting the contributions of different relative time steps, encouraging the model to make precise predictions at current time.

## 3.3 Predictive Multimodal Future Modeling Network

Inspired by advances in online action detection [58, 17, 67], our approach to On-AVEP centers on predictively modeling multimodal pseudo-future sequences using only currently available data. To help PreFM better utilize and consolidate all available modal and temporal cues, we propose a Universal Hybrid Attention (UHA) block to bridge different modalities across time. Given the target query sequence $Q$ and the flexible list of $k$ context sets $\{F_i\}_{i=1}^k$ where each $F_i$ can represent various temporal segments of different modalities, UHA merges these features into $Q$ as follows:

$$\text{UHA}(Q, \{F_i\}_{i=1}^k) = \text{FFN}(\text{LN}(Q + \sum_{i=1}^k \text{Attn}(Q, F_i, F_i))) \tag{1}$$

Where Attn is multi-head attention [57], LN is Layer Normalization, and FFN is a Feed-Forward Network. UHA serves as the foundational attention block for subsequent fusion operations.

**Pseudo-Future Mechanism**   This mechanism first fuses current audio-visual information and then models an initial prediction of the future sequence. Given input features up to current time $T$, $\{(f_t^v, f_t^a)\}_{t=1}^T$, we define a current working window of length $\boldsymbol{L_c}$. This yields the initial current audio and visual features $F_c^a = \{f_t^a\}_{t=T-L_c+1}^T$ and $F_c^v = \{f_t^v\}_{t=T-L_c+1}^T$, both in $\mathbb{R}^{L_c \times D}$.

First, we perform an initial *feature fusion* between $F_c^a$ and $F_c^v$. Each sequence is processed by our UHA block with both as context. The fused current features $\tilde{F}_c^a, \tilde{F}_c^v \in \mathbb{R}^{L_c \times D}$ are produced by:

$$\tilde{F}_c^m = \text{UHA}(F_c^m, \{F_c^a, F_c^v\}), m \in \{a, v\} \tag{2}$$

Next, *future modeling* generates initial pseudo-future sequences of length $\boldsymbol{L_f}$. We use learnable tokens $Q^a, Q^v \in \mathbb{R}^{L_f \times D}$ as queries. These attend to the corresponding fused current features:

$$\tilde{F}_f^m = \text{Attn}(Q^m, \tilde{F}_c^m, \tilde{F}_c^m), m \in \{a, v\} \tag{3}$$

This step yields the initial multimodal pseudo-future sequences $\tilde{F}_f^a, \tilde{F}_f^v \in \mathbb{R}^{L_f \times D}$.

**Temporal-Modality Cross Fusion**   Having obtained the fused current features $(\tilde{F}_c^a, \tilde{F}_c^v)$ and initial pseudo-future sequences $(\tilde{F}_f^a, \tilde{F}_f^v)$, this stage performs further interactions to mutually refine them, reducing potential noise within the pseudo-future, while simultaneously enriching the current representations with foresight gleaned from the modeled future.

First, *future augmentation* refines the initial pseudo-future predictions with UHA block:

$$\hat{F}_f^m = \text{UHA}(\tilde{F}_f^m, \{\tilde{F}_f^a, \tilde{F}_f^v, \tilde{F}_c^m\}), m \in \{a, v\} \tag{4}$$

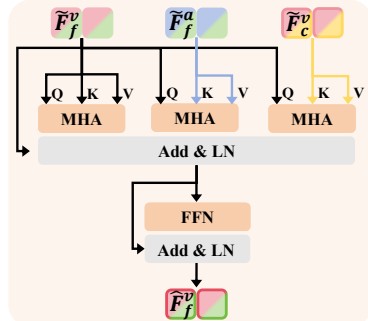

This yields augmented pseudo-future sequences $\hat{F}_f^a, \hat{F}_f^v \in \mathbb{R}^{L_f \times D}$. Notably, the context list within the UHA block enables a rich combination of self-attention, as well as cross-interactions across modalities and time. For instance, the augmented visual pseudo-future $\hat{F}_f^v$ is obtained by interacting with $\tilde{F}_f^v$ (for self-attention), $\tilde{F}_f^a$ (for cross-modal attention), and $\tilde{F}_c^v$ (for cross-temporal attention) as shown in Figure 3.

Figure 3: Temporal-modality cross fusion for the pseudo-future $\hat{F}_f^v$.

Next, *current refinement* integrates the augmented future back into the current representations:

$$\hat{F}_c^m = \text{UHA}(\tilde{F}_c^m, \{\tilde{F}_c^a, \tilde{F}_c^v, \hat{F}_f^m\}), m \in \{a, v\} \tag{5}$$

This results in the final contextually-aware current feature sequences $\hat{F}_c^a, \hat{F}_c^v \in \mathbb{R}^{L_c \times D}$.

Finally, these augmented current and future features are projected by a shared classification head $h(\cdot)$ and a Sigmoid function $\mathcal{S}(\cdot)$ to obtain event predictions $\hat{y}_c \in \mathbb{R}^{L_c \times C}$ (for the current window) and $\hat{y}_f \in \mathbb{R}^{L_f \times C}$ (for the future window):

$$\hat{y}_c = \mathcal{S}(h(\texttt{Concat}(\hat{F}_c^a, \hat{F}_c^v))), \hat{y}_f = \mathcal{S}(h(\texttt{Concat}(\hat{F}_f^a, \hat{F}_f^v))) \tag{6}$$

Here, `Concat` denotes feature concatenation, and $C$ is the event class count (either $C_{av}$ or $C_a + C_v$). For online inference, the prediction in $\hat{y}_c$ corresponding to time $T$ is used. While during training, $\hat{y}_c$ and $\hat{y}_f$ are supervised across the time steps of $[T - L_c + 1, T + L_f]$, using BCE loss with annotations $y_c \in \mathbb{R}^{L_c \times C}$ and $y_f \in \mathbb{R}^{L_f \times C}$:

$$\mathcal{L}_c = \text{BCE}(\hat{y}_c, y_c), \mathcal{L}_f = \text{BCE}(\hat{y}_f, y_f) \tag{7}$$

### 3.4   Modality-agnostic Robust Representation

Learning from the rich, modality-agnostic event representations established by powerful pre-trained teacher models [46, 8, 16, 59, 20] is efficient to obtain robust and generalizable representations

while maintaining efficiency. For each time step $t$, we convert the event labels $y_t$ into "*a/an audio/visual/audio-visual event of* [cls]" as text prompt, which is then processed by the text encoder of frozen teacher model OnePeace [59] to obtain modality-agnostic event features $f_t$. Simultaneously, the student's representation $\hat{f}_t^{av} = \texttt{Concat}(\hat{f}_t^a, \hat{f}_t^v)$ can be easily extracted from our PreFM model, and distilled by the target representation $f_t$. Cosine similarity is used as distillation loss as follows:

$$\mathcal{L}_{mrr} = \frac{1}{L_c + L_f} \sum_{t=T-L_c+1}^{T+L_f} \left(1 - \frac{\hat{f}_t^{av} \cdot h'(f_t)}{\|\hat{f}_t^{av}\| \cdot \|h'(f_t)\|}\right) \tag{8}$$

Where $h'(\cdot)$ denotes the projector module implemented by a linear layer to align different dimensions.

### 3.5 Focal Temporal Prioritization

To emphasize the predictions close to the present moment, we introduce a focal temporal prioritization scheme to the loss calculation, highlighting the significance of prediction at time step $T$ instead of a uniform weighting. Specifically, we define temporal priorities using a Gaussian function centered at the current time $T$: $g(t, \sigma) = \exp\left(-\frac{(t)^2}{2\sigma^2}\right)$, where $t$ is the relative temporal distance from time $T$, and $\sigma$ controls the width of the focus. We define the temporal weights $w_c(t) \in [T - L_c + 1, T]$ for the current window, and $w_f(t) \in [T+1, T+L_f]$ for pseudo-future sequences,

$$w_c(t-T) = \frac{L_c \cdot g(t-T, L_c)}{\sum_{k=T-L_c+1}^{T} g(k-T, L_c)}, \quad w_f(t-T) = \frac{L_f \cdot g(t-T, L_f)}{\sum_{k=T+1}^{T+L_f} g(k-T, L_f)} \tag{9}$$

Let $\mathcal{L}_c(t)$, $\mathcal{L}_f(t)$ and $\mathcal{L}_{mrr}(t)$ be the per-timestep loss in Eq. 7 and Eq. 8. We use $w(t - T) = \texttt{Concat}\{w_c(t - T), w_f(t - T)\}$ to obtain the while weights sequence vector. The final loss is computed as:

$$\mathcal{L} = \sum_{t=T-L_c+1}^{T} w_c(t-T) \cdot \mathcal{L}_c(t) + \sum_{t=T+1}^{T+L_f} w_f(t-T) \cdot \mathcal{L}_f(t) + \lambda \sum_{t=T-L_c+1}^{T+L_f} w(t-T) \cdot \mathcal{L}_{mrr}(t) \tag{10}$$

The hyperparameter $\lambda$ balances the robust representation term (typically 1.0).

### 3.6 Training and Online Inference

**Random Segment Sampling for Training** To adapt training for the online nature of On-AVEP and enhance data utilization, we design a random segment sampling strategy. During training, for a video of total length $T_{all}$, the target prediction times $T_k \in [1, T_{all}]$ are generated by $T_k = kL_c + \delta$. Here, $k$ serves as an index for iterating across the video, and $\delta \in [0, L_c - 1]$ is a periodically selected random integer offset. The $L_c$-length feature sequences $\{(f_t^v, f_t^a)\}_{t=T_k-L_c+1}^{T_k}$ act as model inputs, and zero-padding is applied at the beginning if $T_k < L_c - 1$. This strategy provides diverse training segments with a fixed history length $L_c$, suitable for the online setting.

**Online Inference** During inference, the model works in a truly online manner, processing the input video stream with a sliding window of length $L_c$ and stride 1. At each step $T_{infer}$, the model takes features from $[T_{infer} - L_c + 1, T_{infer}]$, generates the multimodal pseudo-future context, and gets the final event predictions for the current time step $T_{infer}$.

## 4 Experiments

### 4.1 Experimental Setups

**Dataset** **UnAV-100** [13] is a large-scale dataset designed for dense audio-visual event localization in untrimmed videos. It contains 10,790 videos of varying lengths covering 100 event categories, with over 30,000 annotated audio-visual event instances. **LLP** [54] provides 11,849 trimmed 10-second clips across 25 categories for audio-only and visual-only event parsing. For online scenarios, we concatenate LLP clips into longer video sequences. Specifically, half of these sequences are formed

Table 1: Comparison with SOTA methods on On-AVEL task. Feature extractors: *I.V.* denotes I3D [1]+VGGish [24], *O.* denotes OnePeace [59] and *C.C.* denotes CLIP [46]+CLAP [8]. "PreFM+" replaces the feature extractor from CLIP/CLAP to OnePeace and the hidden dimension is expanded from 256 to 512. Methods marked with "*" take the entire video as input, fully utilizing the complete context.

| Methods | Extractors | Segment-Level | | Event-Level | | | | | | Params↓ | FLOPs↓ | Inference | | |
|---|---|---|---|---|---|---|---|---|---|---|---|---|---|---|
| | | F1 | mAP | 0.1 | 0.3 | 0.5 | 0.7 | 0.9 | Avg | | | Memory↓ | FPS↑ | Latency↓ |
| **UnAV** [13] | *I.V.* | 47.5 | 58.3 | 50.9 | 37.1 | 28.7 | 18.2 | 9.4 | 28.6 | 139.4M | 52.4G | 764.7MB | 10.6 | 94.3ms |
| **UniAV** [14] | *O.* | 47.8 | 66.9 | 50.3 | 38.9 | 29.9 | 21.1 | 12.3 | 30.3 | 130.8M | 22.7G | 1020.5MB | 15.6 | 64.1ms |
| **CCNet** [78] | *O.* | 54.8 | 62.3 | 58.3 | 46.3 | 37.5 | 27.3 | 15.8 | 37.0 | 238.8M | 72.1G | 1179.4MB | 7.5 | 133.3ms |
| **PreFM (Ours)** | *C.C.* | 59.1 | 70.1 | 61.5 | 53.6 | 46.9 | 39.6 | 29.2 | 46.3 | 6.5M | 0.4G | 56.4MB | 51.9 | 19.3ms |
| **PreFM+ (Ours)** | *O.* | 62.4 | 70.6 | 66.3 | 58.2 | 52.2 | 44.5 | 35.4 | 51.5 | 13.8M | 0.5G | 144.2MB | 42.0 | 23.8ms |
| UnAV* [13] | *I.V.* | 56.1 | 67.8 | 59.3 | 56.0 | 52.7 | 46.7 | 35.1 | 50.6 | 139.4M | 52.4G | 764.7MB | 10.6 | 94.3ms |
| UniAV* [14] | *O.* | 59.2 | 70.0 | 62.8 | 59.0 | 55.1 | 48.7 | 35.0 | 52.9 | 130.8M | 22.7G | 1020.5MB | 15.6 | 64.1ms |
| CCNet* [78] | *O.* | 65.0 | 70.6 | 69.0 | 65.1 | 61.0 | 53.1 | 40.1 | 58.3 | 238.8M | 72.1G | 1179.4MB | 7.5 | 133.3ms |

Table 2: Comparison with SOTA methods on On-AVVP task. Feature extractors: *R.C.C.* denotes R3D [55]+CLIP [46]+CLAP [8] and *R.R.V.* denotes R3D [55]+ResNet152 [21]+Vsh: VGGish [24]. "PreFM+" increases the hidden dimension from 128 to 256. Methods marked with "*" take the entire 10-second video clips as input, fully utilizing the complete context.

| Methods | Extractors | Segment-Level | | | | | | Event-Level | | | | | | Params↓ | FLOPs↓ | Inference | | |
|---|---|---|---|---|---|---|---|---|---|---|---|---|---|---|---|---|---|---|
| | | $F1_a$ | $F1_v$ | $F1_{av}$ | $mAP_a$ | $mAP_v$ | $mAP_{av}$ | $0.5_a$ | $0.5_v$ | $0.5_{av}$ | $Avg_a$ | $Avg_v$ | $Avg_{av}$ | | | Memory↓ | FPS↑ | Latency↓ |
| **VALOR** [29] | *R.C.C.* | 49.7 | 52.4 | 45.4 | 72.9 | 68.4 | 56.7 | 36.5 | 46.1 | 34.6 | 35.2 | 42.8 | 33.0 | 4.9M | 0.45G | **20.1MB** | 62.2 | 16.1ms |
| **CoLeaF** [49] | *R.R.V.* | 50.7 | 44.5 | 41.0 | 62.8 | 45.8 | 37.3 | 37.9 | 36.4 | 29.6 | 37.3 | 35.5 | 29.7 | 5.7M | 0.25G | 114.1MB | 60.4 | 16.6ms |
| **LEAP** [76] | *R.R.V.* | 50.6 | 49.3 | 45.8 | 73.3 | 64.3 | 54.6 | 40.1 | 42.5 | 35.9 | 38.4 | 39.6 | 34.3 | 52.0M | 1.09G | 204.7MB | 19.3 | 51.8ms |
| **NREP** [27] | *R.C.C.* | 53.7 | 51.4 | 45.5 | 66.5 | 52.7 | 42.3 | 38.9 | 45.6 | 34.2 | 38.3 | 42.3 | 33.5 | 9.6M | 1.69G | 90.2MB | 26.4 | 37.9ms |
| **MM-CSE** [71] | *R.C.C.* | 53.3 | 56.5 | 48.9 | 74.6 | 70.0 | 57.5 | 39.4 | 50.8 | 38.4 | 37.7 | 46.9 | 36.2 | 6.2M | 0.91G | 33.0MB | 36.1 | 27.7ms |
| **PreFM (Ours)** | *R.C.C.* | 60.0 | 59.3 | 53.3 | 80.0 | 73.7 | 61.3 | 47.1 | 50.9 | 42.0 | 46.3 | 50.6 | 41.2 | **3.3M** | **0.22G** | 20.7MB | **94.4** | **10.6ms** |
| **PreFM+ (Ours)** | *R.C.C.* | 61.0 | 60.0 | 54.6 | 80.2 | 73.8 | 61.4 | 48.5 | 51.7 | 43.1 | 47.6 | 51.0 | 42.2 | 12.1M | 0.48G | 55.9MB | 53.5 | 18.7ms |
| VALOR* [29] | *R.C.C.* | 65.6 | 61.8 | 56.5 | 81.4 | 73.7 | 61.4 | 55.1 | 54.9 | 46.7 | 54.0 | 54.2 | 46.0 | 4.9M | 0.45G | 20.1MB | 62.2 | 16.1ms |
| CoLeaF* [49] | *R.R.V.* | 60.5 | 58.0 | 52.4 | 71.7 | 60.7 | 49.3 | 48.3 | 53.0 | 42.1 | 48.7 | 51.8 | 42.5 | 5.7M | 0.25G | 114.1MB | 60.4 | 16.6ms |
| LEAP* [76] | *R.R.V.* | 61.6 | 61.5 | 56.5 | 80.6 | 71.3 | 60.2 | 52.3 | 54.4 | 47.7 | 51.2 | 55.0 | 46.7 | 52.0M | 1.09G | 204.7MB | 19.3 | 51.8ms |
| NREP* [27] | *R.C.C.* | 67.3 | 63.7 | 57.9 | 77.4 | 66.2 | 53.9 | 55.9 | 57.5 | 47.8 | 54.9 | 56.7 | 47.1 | 9.6M | 1.69G | 90.2MB | 26.4 | 37.9ms |
| MM-CSE* [71] | *R.C.C.* | 67.0 | 64.0 | 57.6 | 82.3 | 74.8 | 61.7 | 56.9 | 56.8 | 47.3 | 54.7 | 56.0 | 46.1 | 6.2M | 0.91G | 33.0MB | 36.1 | 27.7ms |

by randomly concatenating clips to simulate the rapid scene variations often encountered in online streaming content; the other half are formed by concatenating clips from the same event category to represent longer, continuous event occurrences. Following recent works [29, 77, 9, 49, 27, 71], segment-wise pseudo labels from CLIP [46, 22] and CLAP [8] are used for supervision.

**Metric** For model performance, we follow prior work [58, 54], using F1-score and mean Average Precision (mAP) as segment-level metrics. For event-level evaluation, consecutive positive segments are treated as a complete event instance. We calculate event-level F1-scores by setting tiou = [0.1:0.1:0.9] [13] and average F1-score (Avg F1-score) for overall performance. For the On-AVVP task, we adhere to the established protocol from VALOR [29], evaluating audio-only (A), visual-only (V), and combined audio-visual (AV, denoted with subscript "av") events. Regarding model efficiency, we assess the number of trainable parameters, FLOPs per inference, peak inference memory and FPS.

**Implementation details** For both tasks, we set 60 training epochs, with the first 10 epochs dedicated to warm-up. A batch size of 128 is used, and AdamW serves as the optimizer with a weight decay of $1e^{-4}$. We set the value $L_c$ of 10 and $L_f$ of 5 as the default setting. CLIP [46] and CLAP [8] are used to extract visual and audio features with a temporal stride set to 1 second, respectively. All experiments are conducted on a single RTX 3090. For the learning rate and the hidden dimension within the attention block, we use $1e^{-3}$ and 256 for On-AVEL, $5e^{-4}$ and 128 for On-AVVP.

## 4.2 Comparison with Existing Work

Our method is benchmarked against recent SOTA methods UnAV [13], UniAV [14] and CCNet [78] on UnAV-100 [13] for the On-AVEL task, while VALOR [29], CoLeaf [49], LEAP [76], NREP [27], and MM-CSE [71] on LLP [54] for the On-AVVP task. We provide two versions of our method: the basic version "PreFM", and the improved version "PreFM+" with larger hidden size.

**Performance Comparison** As shown in Table 1, PreFM clearly achieves new SOTA results for *On-AVEL* task, surpassing the second-best method with significant improvement of **+7.8** in mAP and **+9.3**

in Avg F1-score. Furthermore, our enhanced version, PreFM+, extends these gains to +**8.3** in mAP and +**14.5** in Avg F1-score with only a moderate increase in parameters, highlighting the excellent scalability of the PreFM architecture for applications requiring higher precision. Similarly for *On-AVVP* task shown in Table 2, PreFM demonstrates consistent advantages, achieving improvements of +**3.8** in mAP$_{av}$ and +**5.0** in Avg F1-score$_{av}$ and PreFM+ further elevating performance to +**3.9** in mAP$_{av}$ and +**6.0** in Avg F1-score$_{av}$ over the second-best methods. Notably, we also present the original offline results of these baseline methods (marked with "*") to show their performance under full-context conditions. Even when compared to these results, our online PreFM achieves comparable performance despite predicting with limited context.

These substantial performance gains across both tasks are largely attributed to our core predictive multimodal future modeling (PMFM) design. By dynamically generating and integrating pseudo-future contextual cues from streaming data, PMFM empowers our method to effectively parse environmental states and accurately capture temporal boundaries.

**Efficiency Analysis**    Regarding the *On-AVEL* task (Table 1), PreFM's efficiency is remarkable. PreFM utilizes merely **2.7**% parameters (*6.5M vs 238.8M*) compared to the next best performing method, and it requires only **0.6**% FLOPs (*0.4G vs 72.1G*) and **4.8**% peak memory (*56.4MB vs 1179.4MB*) for a single inference, while running at an impressive 51.9 FPS with merely a latency of **19.3ms**. The compelling efficiency advantage is also evident in the *On-AVVP* task (Table 2). Such ability to deliver SOTA performance with drastically reduced overhead highlights that PreFM is designed with a strong emphasis on practical deployability, rendering it a highly suitable and efficient solution for resource-constrained real-time applications.

### 4.3    Ablation Studies

**Main Component**    To systematically evaluate the contribution of each proposed component, we conduct comprehensive ablation studies on the On-AVEL task, with results presented in Table 3(a). The simple prediction strategy (row 1) uses only data at time $T$ and performs badly. Our baseline (row 2), which just extends accessible data to context $L_c$ but no more improvements, achieves an Avg F1-score of 40.8%. Introducing the pseudo future mechanism ($PF$, row 3) significantly boosts performance to 42.4 (+1.6 vs baseline), underscoring the importance of future context modeling over relying solely on past or current information. Further incorporating modality-agnostic robust representation ($\mathcal{L}_{mrr}$, row 5) or random segment sampling ($RS$, row 6) individually builds upon this, yielding Avg F1-scores of 44.2 (+3.4 vs baseline) and 44.0 (+3.2 vs baseline) respectively, demonstrating their distinct benefits. The focal temporal prioritization ($w(t)$) consistently improves results when applied (e.g., row 4 vs 3, and row 7 vs 5), confirming its effectiveness in focusing the model on critical information at the current moment. Finally, our full PreFM model (row 8), integrating all components, achieves a final Avg F1-score of 46.3, marking a substantial +5.5 improvement over the baseline and validating the collective effectiveness of our design.

**Impact of future-oriented losses**    We investigate the impact of direct future supervision $\mathcal{L}_f$ and future part of robust representation loss $\mathcal{L}_{mrr,f}$ used in pseudo-future ($PF$) mechanism. Results are shown in Table 3(b). A comparison of the first two rows shows that merely incorporating the extra parameters in the future module without applying any future supervision yields negligible performance gains. Conversely, the results in the subsequent three rows indicate that designing losses to explicitly guide the model in anticipating and modeling the future, whether through direct supervision or robust representation distillation, enhances model performance. These findings clearly demonstrate that the performance benefits derived from our pseudo-future mechanism are primarily attributable to the effective learning guided by these targeted future-oriented losses, rather than merely an increase in model capacity.

**Impact of temporal-modality cross fusion**    Generating reliable audio-visual pseudo-future is challenging due to inherent predictive noise. Table 3(c) compares our Temporal-Modality Cross Fusion (TMCF) with ablated variants that utilize only self-attention (Self), audio-visual modality fusion (M only), or temporal-only fusion (T only), focusing on their accuracy in predicting the future (the first three relative time steps) and overall event parsing performance. The inferior performance of these simplified variants underscores that uni-dimensional interactions are insufficient for producing robust future sequences, leaving its reliability and noise levels suboptimal. In contrast, our full TMCF,

Table 3: (a) Overall ablation study. $PF$: the pseudo future mechanism, $w(t)$: focal temporal prioritization, $\mathcal{L}_{mrr}$: modality-agnostic robust representation, $RS$: random segment sampling. (b) Ablation studies for future-oriented losses. (c) Ablation studies for temporal-modality cross fusion. Params: trainable parameters, S-L: Segment-Level, E-L: Event-Level.

| | $PF$ | $w(t)$ | $\mathcal{L}_{mrr}$ | $RS$ | S-L mAP | E-L Avg |
|---|---|---|---|---|---|---|
| (1) | Simple Predictions | | | | 66.6 | 29.9 |
| (2) | ✗ | ✗ | ✗ | ✗ | 69.1 | $40.8_{+0.0}$ |
| (3) | ✓ | ✗ | ✗ | ✗ | 69.7 | $42.4_{+1.6}$ |
| (4) | ✓ | ✓ | ✗ | ✗ | 69.7 | $43.0_{+2.2}$ |
| (5) | ✓ | ✗ | ✓ | ✗ | 69.8 | $44.2_{+3.4}$ |
| (6) | ✓ | ✗ | ✗ | ✓ | 70.5 | $44.0_{+3.2}$ |
| (7) | ✓ | ✓ | ✓ | ✗ | 69.4 | $45.4_{+4.6}$ |
| (8) | ✓ | ✓ | ✓ | ✓ | 70.1 | $\mathbf{46.3}_{+5.5}$ |

(a)

| $\mathcal{L}_f$ | $\mathcal{L}_{mrr,f}$ | $PF$ | Params | S-L mAP | E-L Avg |
|---|---|---|---|---|---|
| ✗ | ✗ | ✗ | 2.6M | 69.8 | $44.7_{+0.0}$ |
| ✗ | ✗ | ✓ | 6.5M | 69.6 | $44.8_{+0.1}$ |
| ✗ | ✓ | ✓ | 6.5M | 69.9 | $45.2_{+0.5}$ |
| ✓ | ✗ | ✓ | 6.5M | 69.6 | $45.5_{+0.8}$ |
| ✓ | ✓ | ✓ | 6.5M | 70.1 | $\mathbf{46.3}_{+1.6}$ |

(b)

| | S-L F1 T+1 | T+2 | T+3 | E-L Avg |
|---|---|---|---|---|
| Self | 55.4 | 54.6 | 53.7 | 44.6 |
| T only | 56.7 | 56.0 | 55.5 | 45.3 |
| M only | 56.6 | 55.7 | 55.1 | 45.0 |
| TMCF | **57.5** | **56.5** | **55.4** | **46.3** |

(c)

Table 4: (a) Ablation studies for different length of past and future. (c) Ablation studies for different pre-trained teacher models.

| | $L_c$ | $L_f$ | Seg-Level F1 | mAP | Event-Level 0.5 | Avg |
|---|---|---|---|---|---|---|
| (1) | 10 | 1 | 58.8 | 70.1 | 46.8 | 45.9 |
| (2) | 10 | 5 | 59.1 | 70.1 | 46.9 | **46.3** |
| (3) | 10 | 10 | 57.3 | 69.9 | 46.4 | 45.4 |
| (4) | 5 | 5 | 57.2 | 69.9 | 44.1 | 43.5 |
| (5) | 20 | 5 | 48.9 | 65.7 | 38.9 | 38.5 |

(a)

| Models | Dimensions | Seg-Level F1 | mAP | Event-Level 0.5 | Avg |
|---|---|---|---|---|---|
| AudioClip [20] | 1024 | 58.7 | **70.2** | 46.6 | 46.2 |
| ImageBind [16] | 1024 | 58.3 | 70.0 | **47.2** | 45.9 |
| ONE-PEACE [59] | 1536 | **59.1** | 70.1 | 46.9 | **46.3** |

(b)

by collaboratively leveraging both cross-modal and cross-temporal interactions from available content, generates more accurate and dependable pseudo-future sequences. This results in a higher-quality predictive context that more effectively mitigates noise and aids robust real-time event parsing.

**Impact of context lengths $L_c$ and $L_f$** We investigate the impact of varying lengths for the working area $L_c$ and the pseudo-future sequence $L_f$, with results presented in Table 4(a). The optimal performance, achieving an Avg F1-score of 46.3, is obtained with our default configuration of $L_c = 10$ and $L_f = 5$ (row 2). Analysis of $L_f$ (rows 1, 2, 3, with $L_c = 10$ fixed) indicates that while a very short future window ($L_f = 1$) provides insufficient predictive insight, an overly long one ($L_f = 10$) can introduce distracting noise, both degrading performance. Similarly, examining $L_c$ (rows 2, 4, 5, with $L_f = 5$ fixed) reveals that too little historical context ($L_c = 5$) offers inadequate support, whereas excessive history ($L_c = 20$) may include outdated or irrelevant information. These findings confirm the importance of appropriately sized context windows, with $L_c = 10$ and $L_f = 5$ providing the most effective balance for the immediate event parsing task.

**Different teacher models in MRR** We evaluate the influence of different pre-trained teacher models on our modality-agnostic robust representation (MRR) module. Specifically, we compare OnePeace [59], ImageBind [16], and AudioClip [20] across multiple metrics. As the results demonstrate in Table 4(b), no single model consistently outperforms the others on every measure. However, OnePeace delivers better segment-level F1-scores and average event-level performance, which lead us to adopt it as our default teacher.

**Temporal impact of the pseudo-future** Figure 4(a) illustrates how our pseudo future ($PF$) mechanism affects prediction accuracy across time steps relative to the current moment $T$. From the orange line, we observe that the model's peak performance occurs significantly earlier (around relative time $T - 6$), with accuracy declining as it approaches $T$, indicating a strong reliance on full context. In contrast, the purple line shows that incorporating the $PF$ not only achieves generally higher accuracy but also shifts its performance peak much closer to the actual target time $T$ (around $T - 2$). These observations underscore a fundamental principle in event parsing: accurate event identification intrinsically depends on a comprehensive contextual window. Thus, the reliance on future context presents a significant hurdle for online systems. Our $PF$ mechanism effectively

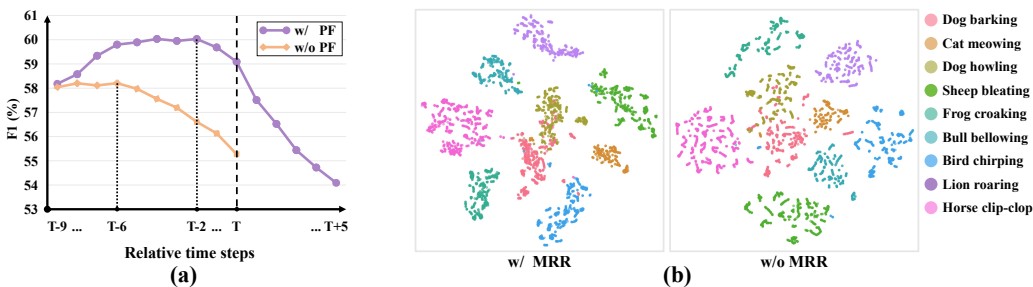

Figure 4: (a) The performance across different relative time steps. (b) t-SNE visualization of the pre-classifier features. We use nine animal events from UnAV-100 [13] for better illustration.

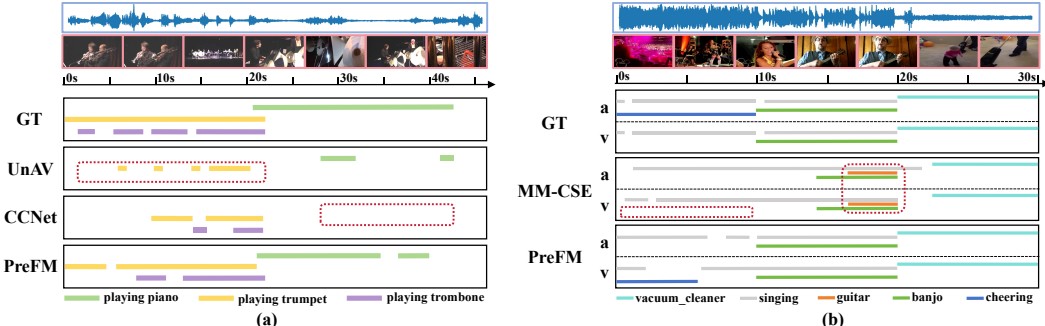

Figure 5: (a) The visualization on the On-AVEL task. (b) The visualization on the On-AVVP task. GT: ground truth. The red dotted box indicates the area of mispredictions.

anticipates event trends, models and utilizes audio-visual future information to enhance prediction accuracy near the present moment, thereby mitigating immediate contextual limitations.

**Impact of MRR on latent feature representation**  Figure 4(b) qualitatively evaluates our modality-agnostic robust representation (MRR) via t-SNE visualization of latent features from nine predefined animal events from UnAV-100 [13]. With MRR, event classes form more compact and well-separated clusters, unlike the more chaotic clusters from the model without MRR. This suggests that while MRR may shift the latent space, it guides the model towards more discriminative representations, enhancing event separability and overall performance.

### 4.4 Qualitative Analysis

Figure 5 presents a qualitative comparison of our PreFM with SOTA methods UnAV [13], CCNet [78] on On-AVEL task and MM-CSE [71] on On-AVVP tasks. Prior methods often exhibit limitations such as missed detections (e.g., "trombone" by UnAV, "piano" by CCNet, "cheering" by MM-CSE), fragmented predictions (e.g., "trumpet" by UnAV) depicted by red dotted box. In stark contrast, PreFM's predictions exhibit strong temporal continuity and precise event boundary localization, without the interruptions or errors in other methods. These visualizations intuitively showcase PreFM's enhanced recognition accuracy and the coherent, continuous nature of its event parsing.

## 5   Conclusions

In this work, we introduce online audio-visual event parsing to enable real-time multimodal event understanding in streaming videos. We identify accurate online inference and real-time efficiency as two crucial capabilities in this setting, and propose the PreFM framework, featuring a novel predictive multimodal future modeling to infer future context and modality-agnostic robust representation together with focal temporal prioritization for model's generalization. Extensive experiments on the UnAV-100 and LLP datasets validate that PreFM significantly outperforms prior methods, achieving state-of-the-art performance while offering a superior balance between accuracy and computational efficiency, thus presenting a viable solution for practical real-time multimodal applications.

## Acknowledgments and Disclosure of Funding

This work is supported by the National Natural Science Foundation of China (No. 92470203, U23A20314), the Beijing Natural Science Foundation (No. L242022), and the Fundamental Research Funds for the Central Universities (No. 2024XKRC082).

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

# A Technical Appendices and Supplementary Material

## A.1 Limitations and Broader Impact

**Limitations**  While PreFM demonstrates promising results in online audio-visual event parsing, we identify a couple of key avenues for future exploration and enhancement. Firstly, the current PreFM design is primarily tailored for event detection and localization. Further research can extend its capabilities to more complex, semantically rich tasks such as video question answering or detailed captioning, and enhance its capacity for long-range temporal reasoning, potentially through integration with large language models. Secondly, while PreFM's predictive modeling of pseudo-future context is a key component for enhancing online inference, the inherent nature of future prediction means that the generated cues may not always perfectly foresee subsequent events. Although our temporal-modality cross fusion (TMCF) (detailed in Sec. 3.3) is designed to refine these predictions and mitigate potential noise by leveraging cross-modal and cross-temporal interactions—with its positive impact analyzed in Sec. 4.3— the noise may somewhat degrade performance. While TMCF offers an initial solution, further research can be developed to enhance the reliability and effectiveness of the future.

**Broader Impact**  Our work on online audio-visual event parsing, using methods like PreFM, can greatly improve real-time AI systems. However, we must think carefully about serious ethical issues when using audio and video data. Important issues include protecting people's privacy from unwanted watching or access, reducing unfair biases that the AI might learn from its data should be considered.

## A.2 Additional Results and Analysis

**Quantitative Analysis of the Pseudo-Future's Quality**  We evaluate them from two perspectives, their semantic similarity to ground-truth event features, and their effectiveness in predicting future events: *Top-k Similarity Accuracy*: We measure if a generated feature vector at a relative future time step (T+1-T+5) is the Top-1 or Top-5 closest match to its corresponding ground-truth class feature embedding, among all 100 classes in UnAV-100. *Future Prediction F1-Score*: We also report the standard segment-level F1-score for predictions made for future time steps. The results are presented in the Table 5.

This results provides two key insights: First, the pseudo-future features are remarkably realistic. The Top-5 Similarity Accuracy of over 94% demonstrates that the correct event feature is almost always ranked among the top candidates. Second, these high-quality features enable strong future prediction performance, as evidenced by the solid F1-scores.

Table 5: The quantitative results of the pseudo-future's quality.

| Metric | T+1 | T+2 | T+3 | T+4 | T+5 |
|---|---|---|---|---|---|
| Top-1 Similarity | 45.4 | 44.6 | 44.0 | 43.3 | 42.6 |
| Top-5 Similarity | 95.5 | 95.3 | 95.1 | 94.8 | 94.3 |
| F1 | 57.5 | 56.5 | 55.4 | 54.7 | 54.1 |

**Ablation study on hyperparams**  The ablation study on the loss weighting hyperparameter $\lambda$ is shown in Table 6. The results indicate that performance diminishes at the tested extreme values ($\lambda = 0.1$ and $\lambda = 10$), while the model exhibits stable and strong performance across a moderate range (from $\lambda = 0.5$ to $\lambda = 2$). Therefore, we adopt $\lambda = 1$ as the default setting in our experiments for simplicity.

Table 6: Ablation studies for hyperparams, loss weight $\lambda$.

| $\lambda$ | Seg-Level | | Event-Level | |
|---|---|---|---|---|
| | F1 | mAP | 0.5 | Avg |
| 0.1 | 58.3 | 70.8 | 46.9 | 45.8 |
| 0.5 | 58.9 | 70.2 | 47.2 | 46.2 |
| 1 | 59.1 | 70.1 | 46.9 | 46.3 |
| 1.5 | 58.4 | 69.9 | 47.2 | 46.2 |
| 2 | 59.2 | 70.0 | 47.5 | 46.6 |
| 5 | 55.9 | 68.9 | 44.3 | 43.7 |
| 10 | 13.2 | 50.8 | 9.4 | 9.5 |

**Different feature extractors**   Table 7 presents the ablation study on different feature extractors, evaluating their impact on the performance and efficiency of On-AVEP task. The results clearly indicate that employing more powerful foundation models as feature extractors generally leads to significant improvements in parsing performance. Specifically, while the I3D [1]+VGGish [24] combination is relatively lightweight, its performance is comparatively limited. In contrast, AudioClip [20] and CLIP [46]+CLAP [8] offer a favorable balance between performance and computational efficiency. Although OnePeace [59] achieves the best parsing results, its substantial computational requirements may hinder its practical deployment in real-world scenarios. Notably, the computational complexity of our proposed PreFM module remain relatively stable and low across all tested feature extractors. This underscores that the feature extraction stage constitutes the primary performance bottleneck and source of computational load, directly impacting the system's online processing capabilities.

Table 7: Ablation studies for different feature extractors. a: audio extractor part, v: visual extractor part.

| Methods | Seg-Level | | Event-Level | | Dimensions | | FLOPS↓ | | |
|---|---|---|---|---|---|---|---|---|---|
| | F1 | mAP | 0.5 | Avg | a | v | a | v | PreFM |
| I3D [1]+VGGish [24] | 30.7 | 48.7 | 23.7 | 24.0 | 128 | 2048 | 0.9G | 3.5G | 0.3G |
| AudioClip [20] | 48.0 | 63.6 | 37.7 | 37.0 | 1024 | 1024 | 2.7G | 5.4G | 0.1G |
| CLIP [46]+CLAP [8] | 59.1 | 70.1 | 46.9 | 46.3 | 768 | 768 | 23.1G | 77.8G | 0.5G |
| ONE-PEACE [59] | 62.4 | 70.6 | 52.2 | 51.5 | 1536 | 1536 | 78.8G | 389.8G | 0.4G |

**Failure cases**   The quantitative findings and analysis about failure cases are shown below:

*Confusion Between Similar Events*   We analyze the events with the lowest performance and their most common confusions in Table 8. This results reveal that PreFM struggles to distinguish between events that are semantically or acoustically similar. We hypothesize this is because our current framework does not explicitly incorporate a contrastive learning design to better separate the representations of events originating from similar audio-visual sources.

Table 8: The quantitative analysis of confusion between similar events.

| Event | Precision | Recall | F1 | Most confused with |
|---|---|---|---|---|
| People slurping | 0.55 | 0.17 | 0.25 | People eating, man speaking |
| People shouting | 0.42 | 0.19 | 0.27 | Baby laughter, engine knocking |

*Performance in Dense Scenes*   We analyze the impact of event density (the number of event classes within a video) on event-level performance in Table 9. These results show that PreFM's performance degrade in complex videos containing a large number of distinct event classes. This suggests that while our future modeling is effective, its benefits are less pronounced in scenarios with very rapid scene changes and drastic context shifts.

Table 9: The quantitative analysis of PreFM in dense scenes.

| Num events | Event-Level Avg |
|---|---|
| 1-3 | 0.54 |
| 4-6 | 0.23 |
| >6 | 0.13 |

## A.3 Additional Details

**The detailed process of modality-agnostic representation refinement (MRR)** For the MRR process, we select OnePeace [59] as the pre-trained teacher model. The generation of target teacher features $f_t$ at each time step $t$ involves the following steps: First, ground-truth event labels $y_t$ are converted into textual prompts using the template "*a/an audio/visual/audio-visual event of* [cls]". These prompts are then processed by the OnePeace text encoder to yield the modality-agnostic event features. If multiple events are active at time $t$, the final $f_t$ is computed by averaging the features corresponding to all active event classes. Concurrently, the student model's representation $\hat{f}_t^{av}$ is prepared. We extract the audio features $\hat{f}_t^a$ and visual features $\hat{f}_t^v$ from our model at time $t$, specifically from the layer before the final classification head $h(\cdot)$. These extracted features are subsequently concatenated in feature dimension to form the student's representation: $\hat{f}_t^{av} = \text{Concat}(\hat{f}_t^a, \hat{f}_t^v)$.

**Specific values for focal temporal prioritization** As detailed in Eq. 9 and Eq. 10, our focal temporal prioritization are designed to emphasize predictions closer to the current time $T$ while maintaining the overall loss scale. This scale preservation ensures that the sum of weights for the context window, $\sum_{t=T-L_c+1}^{T} w_c(t)$, equals $L_c$, and for the future window, $\sum_{t=T+1}^{T+L_f} w_f(t)$, equals $L_f$. Table 10 presents the specific numerical values of these weights for each relative time step, calculated with our default settings of $L_c = 10$ and $L_f = 5$.

Table 10: Specific values of the focal temporal prioritization for current time steps ($t \in [T - 9, T]$) and future prediction time steps ($t \in [T + 1, T + 5]$).

| Time step | Current | | | | | | | | | | Future | | | | |
|---|---|---|---|---|---|---|---|---|---|---|---|---|---|---|---|
| | T-9 | T-8 | T-7 | T-6 | T-5 | T-4 | T-3 | T-2 | T-1 | T | T+1 | T+2 | T+3 | T+4 | T+5 |
| Weight value | 0.76 | 0.83 | 0.89 | 0.95 | 1.01 | 1.06 | 1.09 | 1.12 | 1.14 | 1.14 | 1.12 | 1.10 | 1.03 | 0.94 | 0.81 |

**Difference between random segment sampling and normal sampling** The details of our random segment sampling strategy are described in Sec. 3.6. In contrast, normal sampling strategy just involves dividing a video of total length $T_{all}$ into a sequence of $k$ non-overlapping chunks, each of length $L_c$. For such a normal approach, the target prediction time $T_k$ for each $k$-th chunk is deterministically set to its final time step, specifically defined as $T_k = kL_c - 1$. This means that predictions are consistently targeted only at the very end of these fixed chunks, unlike the more varied and diverse target prediction times generated by our random segment sampling method.

**Dataset modification for online setting** For the **UnAV-100** dataset [13], while its original annotations specify continuous time segments for events (e.g.,$[cls, T_{start}, T_{end}]$), we convert these into frame-level discrete labels for our online task. Specifically, for any given time $T$ and a particular event within a video stream, a label of 1 indicates the event is currently occurring, while 0 indicates it is not.

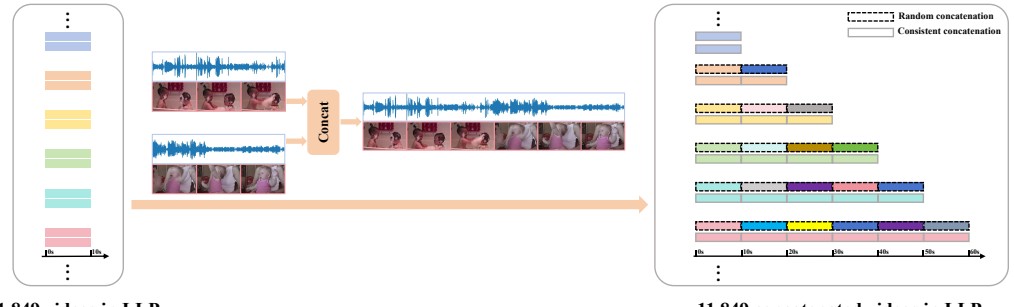

**11,849 videos in LLP**  **11,849 concatenated videos in LLP**

Figure 6: Visualization of the LLP dataset adaptation for the online setting. Various colored rectangles represent video clips of different categories. Video clips are processed by random concatenation and consistent concatenation.

The **LLP** dataset [54] initially provides 11,849 10-second video clips. To adapt it for our online evaluation setting, we concatenate 11,849 new, untrimmed video streams as shown in Figure 6. Each new stream is created by using the original 10-second clips as its base and concatenating additional clips. These resulting streams are specifically constructed to achieve one of six distinct target total durations: *10 seconds* (representing the original clip itself), *20 seconds, 30 seconds, 40 seconds, 50 seconds, or 60 seconds*. Approximately an equal number of streams are generated for each of these six target durations.

This concatenation process employs two distinct strategies: half of the 11,849 streams are formed by *random concatenation*, randomly combining clips from different categories. This aims to simulate the rapid and complex within-second scene variations commonly observed in current streaming content. The other half are constructed by *consistent concatenation*, identifying the first event category present in the base clip and then concatenating multiple additional clips that also contain this specific event, thereby simulating longer videos with a consistent, ongoing event context. This approach allows us to assess the model's adaptability to complex dynamic scenes and its capability for consistent understanding and discrimination within extended event contexts.

**Regarding other datasets: The case of LFAV**   The LFAV dataset [25], comprising 5175 untrimmed videos with diverse audio, visual, and audio-visual events, is designed for long-form audio-visual video parsing and thus appears initially relevant for the On-AVEP task. However, we identify two critical limitations that preclude its effective use with our PreFM framework.

Firstly, complete access to the original video data is restricted. Of the officially stated 3721 training, 486 validation, and 968 test samples, our attempts allow us to retrieve only 3512, 461, and 910 samples respectively (totaling 4883 out of 5175 raw videos). The LFAV benchmark provides pre-extracted features using VGGish [24], ResNet18 [21], and R3D [55]. This reliance on fixed features prevents us from employing different feature extractors or leveraging pre-trained models (such as OnePeace [59] for our modality-agnostic robust representation) directly on the raw video data, which is a key aspect of our method.

Secondly, LFAV is curated under a weak supervision paradigm, offering only video-level annotations for its training set. The absence of readily available segment-level ground truth makes LFAV unsuitable for training critical components of our PreFM model. Specifically, mechanisms like our predictive future modeling and focal temporal prioritization require finer-grained temporal supervision than what LFAV's training annotations provide, rendering it incompatible with the training requirements for our online streaming prediction approach.

**More online inference details**   All methods are evaluated using their officially provided checkpoints; for those without an available checkpoint, we reproduce the results using their official code. For any prediction at time $T$ in online testing, only data from $0$ to $T$ is available.

For **On-AVEL** tasks, SOTA methods [13, 14, 78] pad the entire video beyond $T + 1$ with zeros as input because these methods originally utilize the complete video, and we use this padding to ensure a uniform input length under online settings. Our method, in contrast, does not use all available

historical data up to $T$; instead, it processes the segment $[T - L_c + 1, T]$ as input to derive the prediction at time $T$.

For **On-AVVP** tasks, SOTA approaches [29, 49, 76, 27, 71] employ the segment $[T - 9, T]$ as input, since these methods are designed for 10-second video clips. Similarly, our method utilizes the segment $[T - L_c + 1, T]$ as input for making predictions at time $T$.

**Re-evaluation of efficiency metrics for fair comparison**    Table 11 (for the On-AVEL task) and Table 12 (for the On-AVVP task) present side-by-side comparisons of efficiency metrics. These include figures from our standardized re-evaluation (denoted as "Our Eval.") and those originally published in the respective papers (denoted as "Reported").

For our evaluations, we adhere to a strict and consistent protocol. The number of trainable parameters for all models is calculated as the sum of elements in all parameters requiring gradients (using `sum(p.numel() for p in model.parameters() if p.requires_grad)`). To measure FLOPs, we consistently employ the `thop` library for all methods, assessing a single forward pass (via `flops, _ = profile(model, inputs=(input,))`). All our efficiency tests are conducted under identical environmental conditions to ensure reproducibility and a fair basis for comparison.

Discrepancies may be observed between our "Our Eval." figures and the "Reported" values from the original publications. Such differences can arise from variations in measurement methodologies, specific versions of libraries used, or the underlying hardware and software environments. We present both sets of values to offer a transparent perspective, respecting the data from original publications while providing a benchmark that is directly comparable across methods under our unified testing framework.

Table 11: Comparison of re-evaluated ("Our Eval.") and originally reported ("Reported") efficiency metrics on the On-AVEL task. ("-" indicates the metric was not provided in the original paper.

| Methods | Params | | FLOPs | |
|---|---|---|---|---|
| | Our Eval. | Reported | Our Eval. | Reported |
| UnAV [13] (CVPR2023) | 139.4M | - | 52.4G | - |
| UniAV [14] (Arxiv2404) | 130.8M | 130M | 22.7G | - |
| CCNet [78] (AAAI2025) | 238.8M | - | 72.1G | - |
| PreFM | 12.3M | none | 0.4G | none |
| PreFM+ | 36.9M | none | 0.5G | none |

Table 12: Comparison of re-evaluated ("Our Eval.") and originally reported ("Reported") efficiency metrics on the On-AVVP task. "-" indicates the metric was not provided in the original paper.

| Methods | Params | | FLOPs | |
|---|---|---|---|---|
| | Our Eval. | Reported | Our Eval. | Reported |
| VALOR [29] (NeurIPS2023) | 4.9M | 5.1M | 0.45G | 0.45G |
| Coleaf [49] (ECCV2024) | 5.7M | - | 0.25G | 48.2G |
| LEAP [76] (ECCV2024) | 52.0M | 52.0M | 1.09G | 0.79G |
| NREP [27] (TNNLS2024) | 9.6M | 9.6M | 1.69G | 0.37G |
| MM-CSE [71] (AAAI2025) | 6.2M | 4.5M | 0.91G | 0.80G |
| PreFM | 3.3M | none | 0.22G | none |
| PreFM+ | 12.1M | none | 0.48G | none |

## A.4    More Qualitative Analysis

**On-AVEL**    Figure 7 provides further qualitative validation of our method's on the On-AVEL task through four distinct examples, comparing our results ("Ours") against the state-of-the-art CCNet [78] model ("SOTA"). These visualizations collectively demonstrate that our approach consistently yields event localizations that are more aligned with the ground truth annotations compared to CCNet.

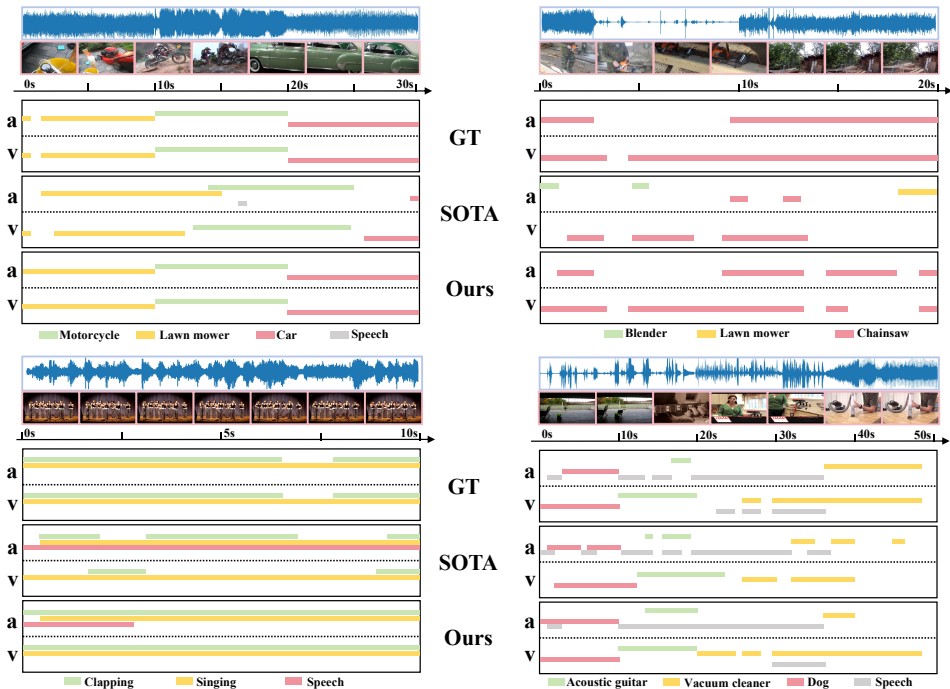

Figure 8: More visualization results on the On-AVVP task.

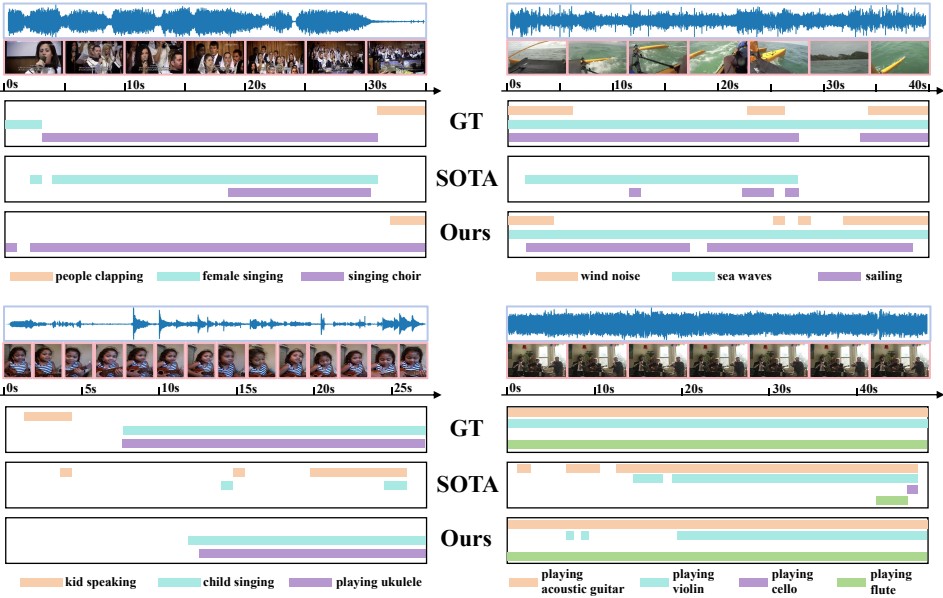

Figure 7: More visualization results on the On-AVEL task.

**On-AVVP** Similarly, we provide further qualitative results of our method's on the On-AVVP task through four distinct examples, comparing our results ("Ours") against the state-of-the-art MM-CSE [71] model ("SOTA"). The visualization results are shown in Figure 8. These visualizations highlight our method's superior performance in precisely parsing events and reducing errors compared to the SOTA model.

This robust handling of both unimodal and multimodal event characteristics signifies a key advantage of our approach for the online audio-visual event parsing task.

