# OpenReview forum: "PreFM: Online Audio-Visual Event Parsing via Predictive Future Modeling"
_NeurIPS.cc/2025/Conference — NeurIPS 2025 poster_

### Official Review · Reviewer_jUXg · 2025-06-24

**Clarity:** 3
**Significance:** 3
**Originality:** 2
**Rating:** 4
**Confidence:** 4

**Summary:**

This paper proposes PreFM, a framework for online audio-visual event parsing. It introduces predictive future modeling, modality-agnostic representation, and temporal weighting to improve real-time performance. The method is evaluated on UnAV-100 and LLP, showing strong accuracy and high efficiency compared to prior work.

**Questions:**

1. How does the model maintain online consistency when it uses future-supervised signals during training?

2. Can you provide a quantitative analysis of the pseudo-future features’ quality?

3. How sensitive is the model to the choice of teacher in MRR?

4. The modality-agnostic distillation depends on OnePeace as the teacher model. Have you tried other options such as CLIP or AudioCLIP? How much does the downstream performance vary?

5. How does PreFM perform in a real streaming setting with long, continuous videos?

**Ethical Concerns:**

["NO or VERY MINOR ethics concerns only"]

**Final Justification:**

I believe the paper has merit in its current form, and while there may be areas for improvement, I maintain my score based on its overall contribution.

**Limitations:**

Key issues like the train-test mismatch and generalization to real-world streaming settings are not discussed.

**Paper Formatting Concerns:**

No issue observed.

**Quality:**

3

**Strengths And Weaknesses:**

**Strengths:**

1. The paper addresses a practical and timely problem: real-time event understanding from streaming multimodal input.

2. The model achieves significant performance improvements over strong baselines while using much fewer parameters.

**Weaknesses:**

1. The model uses future labels and supervision during training, which breaks the online assumption and creates a train-test mismatch.

2. The paper does not evaluate whether the pseudo-future features are realistic or actually useful during inference.

3. The distillation method depends heavily on a specific teacher model, but the paper does not analyze its effect or generalization.

4. The evaluation setting constructs artificial video streams by clip concatenation, which does not reflect real-world online deployment.

---

> ### Author Rebuttal · Authors · 2025-07-31
>
> Thank you for your positive feedback and constructive comments. In regard to the weaknesses and questions raised, we provide our point-to-point responses below:
>
> >**W1 and Q1**:
> Online assumption and train-test mismatch: How does the model maintain online consistency when it uses future-supervised signals during training?
>
> **A1**:
>
> Our framework strictly adheres to the online setting during inference. The use of future labels is a deliberate design choice confined to the training phase, and it does not create a train-test mismatch.
>
> 1. **Strict Adherence to Online Assumptions and Consistency During Reasoning**: As detailed in Sec 3.6, at any given time step $T$, the model's prediction relies solely on historical and current features available up to $T$. No future ground truth or features are accessible, ensuring true online consistency at test time.
> 2. **A Deliberate and Principled Training Strategy**: The use of future labels during training is an explicitly designed strategy to equip the model with the ability to anticipate the future, which is the core of our Predictive Future Modeling (PreFM) framework. Our goal is to have the model internalize the predictive capability.
>
> In summary, this design choice **does not introduce a train-test mismatch and serves as a necessary supervision mechanism**. It effectively bridges the gap by teaching the model to generate its own useful context at inference time, addressing the inherent challenge of missing future information in an online setting.
>
> ---
>
> >**W2 and Q2**:
> The quality and usefulness of pseudo-future features: Can you provide a quantitative analysis of the pseudo-future features’ quality?
>
> **A2**:
>
> Here we provide the analysis of the quality and usefulness of pseudo-future features:
>
> **1. Quantitative Analysis of the Pseudo-Future's Quality**
>
> We evaluate them from two perspectives, their semantic similarity to ground-truth event features, and their effectiveness in predicting future events:
>
> * Top-k Similarity Accuracy: We measure if a generated feature vector at a relative future time step (T+1~T+5) is the Top-1 or Top-5 closest match to its corresponding ground-truth class feature embedding, among all 100 classes in UnAV-100.
> * Future Prediction F1-Score: We also report the standard segment-level F1-score for predictions made for future time steps.
>
> The results are presented in the table below:
>
> | Metric| T+1  | T+2  | T+3 | T+4 | T+5 |
> |-|---|---------|-----|-----|-----|
> |Top-1 Similarity | 45.4 | 44.6 | 44.0  | 43.3| 42.6|
> |Top-5 Similarity | 95.5 | 95.3 | 95.1| 94.8| 94.3|
> |F1 Score|57.5|56.5|55.4|54.7|54.1|
>
> This results provides two key insights:
> * First, the pseudo-future features are remarkably realistic. The Top-5 Similarity Accuracy of over 94% demonstrates that the **correct event feature is almost always ranked among the top candidates**.
> * Second, these **high-quality features enable strong future prediction performance**, as evidenced by the solid F1-scores.
>
> **2. Quantitative Analysis of the Pseudo-Future's Usefulness for Inference**
>
> We now demonstrate their usefulness for the main task—improving prediction at the current time $T$. Our ablation study on the Predictive Future ($PF$) mechanism in Tab.3(b) in our manuscript directly addresses this. For the reviewer's convenience, we highlight the key comparison below:
>
> | $\mathcal L_f$ | $\mathcal L_{mrr,f}$ | $PF$ | mAP | Avg F1|
> | -------- | -------- | -------- |-|-|
> | ×     | ×     | ×     |69.8|44.7|
> |√|√|√|**70.1**|**46.3 (+1.6)**|
>
> The results clearly show that incorporating the $PF$ mechanism, which **generates and utilizes these high-quality future features, leads to a significant performance improvement** (e.g., +1.6 in Avg F1 Score). This quantitatively proves that the pseudo-future context is actually useful during inference for enhancing the model's understanding of the present moment.
>
> In conclusion, our comprehensive analyses confirm that the pseudo-future features are both realistic in their semantic representation and highly useful for boosting the model's final online parsing performance.
>
> ---
>
> >**W3, Q3 and Q4**:
> The choice of teacher model: How sensitive is the model to the choice of teacher in MRR? Have you tried other options such as CLIP or AudioCLIP? How much does the downstream performance vary?
>
> **A3**:
>
> **PreFM's performance is robust and not overly sensitive to the choice of the teacher model.**
>
> We include this analysis in our original submission in Tab.5 in the appendix. For the reviewer's convenience, we present the key results below:
>
> | Teacher Models       | Dimensions  | mAP | Avg F1 |
> |--|--|---|-|
> | AudioClip    | 1024 | 70.2  | 46.2  |
> | ImageBind    | 1024  | 70.0    | 45.9    |
> | ONE-PEACE    | 1536   | 70.1       | 46.3      |
>
> As the table shows, our PreFM framework demonstrates robust and stable performance across different powerful, pre-trained teacher models. The variation in the final Event-Level Avg F1-score is minor (ranging from 45.9 to 46.3).
>
> This indicates that while the choice of teacher has a slight impact, our MRR approach **is not overly sensitive to a specific teacher and generalizes well to other strong multimodal teacher models**.
>
> Regarding the reviewer's suggestion to try CLIP: we specifically chose teacher models like AudioClip, ImageBind, and OnePeace because they provide a unified embedding space that includes the audio modality. Standard CLIP models are primarily designed for vision-language tasks and **do not process audio**, making it unsuitable for providing modality-agnostic supervision for our audio-visual parsing task.
>
> ---
>
> >**W4 and Q5**:
> Evaluation with clip concatenation: How does PreFM perform in a real streaming setting with long, continuous videos?
>
> **A4**:
>
> Our primary evaluation is conducted on the UnAV-100 dataset, which consists of long, untrimmed, in-the-wild videos and directly addresses this concern. The LLP dataset acts as a complementary benchmark.
>
> **1. SOTA Performance on a Real-World Continuous Video Dataset (UnAV-100)**
>
> The UnAV-100 dataset contains 10,790 videos of varying length and it is specifically designed for dense event localization in **long, untrimmed, real-world videos**. Our SOTA results on UnAV-100, detailed in Tab.1 of our manuscript, provide strong evidence of PreFM's effectiveness. For convenience, we show the key results below and include the latency results:
>
> |On-AVEL|mAP↑|Avg F1↑|Params↓|FLOPs↓|Peak Memory↓|FPS↑|Latency↓|
> |-|-|-|-|-|-|-|-|
> |UnAV|58.3|28.6|139.4M|52.4G|764.7MB|10.6|94.3ms|
> |UniAV|66.9|30.3|130.8M|22.7G|1020.5MB|15.6|64.1ms|
> |CCNet|62.3|37.0|238.8M|72.1G|1179.4MB|7.5|133.3ms|
> |PreFM (Ours)|70.1|46.3|**6.5M**|**0.4G**|**56.4MB**|**51.9**|**19.3ms**|
> |PreFM+ (Ours)|**70.6**|**51.5**|13.8M|0.5G|144.2MB|42.0|23.8ms|
>
> As the results show, PreFM significantly outperforms all SOTA methods on this challenging dataset (e.g., achieving a **+9.3** gain in event-level Avg F1-score over the next best method) while using significantly fewer parameters and lower computational complexity. This demonstrates its strong performance and robustness in realistic, continuous video streams.
>
> **2. The Role of the LLP Dataset as a Complementary Benchmark**
>
> We designe the modified LLP dataset  not as a substitute for real-world data, but as a controlled, complementary benchmark to analyze model capabilities under two distinct and challenging online scenarios:
>
> * Random Concatenation: This setup simulates **the rapid and unpredictable scene changes** common in online streaming content, testing the model's adaptability.
> * Consistent Concatenation: This setup simulates **longer, continuous event occurrences**, testing the model's ability for consistent understanding and discrimination over an extended context.
>
> This approach allows us to perform a more nuanced assessment of PreFM's robustness, complementing the evaluation on the in-the-wild UnAV-100 dataset.
>
> **3. Qualitative Visualization on Continuous Streams**
>
> We provide extensive qualitative results in Fig.5 (main paper) and Fig.7 & 8 (appendix). These figures visualize PreFM's superior ability to generate accurate and temporally coherent event boundaries on long video timelines from both datasets.
>
> In summary, PreFM's effectiveness in realistic streaming settings has been thoroughly validated through its SOTA quantitative performance on the large-scale UnAV-100 dataset, supported by targeted analyses on our complementary LLP benchmark and extensive qualitative evidence.

---

> > ### Comment · Reviewer_jUXg · 2025-08-06
> > **Response  to Authors**
> >
> > I would like to sincerely thank the authors for the detailed rebuttal and clarifications. My main concerns regarding the online assumption, pseudo-future feature usefulness, and teacher model sensitivity have been addressed.
> >
> > Minor limitations remain regarding real-world streaming robustness, but they do not affect my overall borderline-accept recommendation. Overall, I find the contribution solid and I support its acceptance.

---

### Official Review · Reviewer_No1a · 2025-06-27

**Clarity:** 3
**Significance:** 4
**Originality:** 3
**Rating:** 5
**Confidence:** 5

**Summary:**

This paper proposes a new task, Online Audio-Visual Event Parsing (On-AVEP), which aims to detect audio and visual events in a streaming setting using only past and current information. To tackle the challenges of limited context and real-time efficiency, the authors introduce PreFM, a novel framework built on three key components: (1) Predictive Multimodal Future Modeling (PMFM) to synthesize pseudo-future cues, (2) Modality-agnostic Robust Representation (MRR) via distillation from a frozen teacher, and (3) Focal Temporal Prioritization to emphasize current-time predictions. Extensive experiments on UnAV-100 and LLP datasets demonstrate that PreFM achieves state-of-the-art performance while being highly efficient in terms of parameters and computation.

**Questions:**

Please refer to the questions mentioned in 'weaknesses' section.

**Ethical Concerns:**

["NO or VERY MINOR ethics concerns only"]

**Final Justification:**

The authors have addressed my questions thoroughly in the rebuttal. I have also considered the insightful comments from other knowledgeable reviewers. The additional experiments and detailed explanations provided in the response adequately resolve the major concerns. This paper proposes a novel and timely task, presents an effective and efficient baseline, and includes extensive experiments accompanied by comprehensive discussions. Overall, I believe this is a strong submission and deserving of acceptance. I maintain my original positive rating.

**Limitations:**

Yes.

**Quality:**

4

**Strengths And Weaknesses:**

## Strengths

- **Novel Task Definition:** The formulation of On-AVEP as a real-time, streaming version of AVEP is timely and practically relevant, filling a gap in existing literature that mainly focuses on offline processing.
- **Strong Efficiency-Performance Trade-off:** Despite its lightweight architecture (e.g., 6.5M parameters), PreFM significantly outperforms prior methods with much larger models, making it highly suitable for deployment in real-time scenarios.
- **Thorough Empirical Evaluation:** The paper provides comprehensive comparisons, ablations, and qualitative analyses, making the improvements clearly attributable to each design component.
- **Good Clarity and Writing:** The paper is well-organized and easy to follow, with clear figures and sound motivation for each component.



## Weaknesses

Given the aforementioned strengths, I am generally satisfied with this paper and would like to share a few questions and suggestions:

- **Contradictory statements?** Lines 256–257 suggest that the PF module significantly boosts performance, whereas Lines 268–269 state that PF yields negligible gains. It would be helpful to clarify these descriptions to avoid confusion.

- **Failure cases.** While Figure 5 and the supplementary materials present visualizations, the paper would benefit from a more detailed discussion of the failure cases—when and why PreFM might not perform well.

- **Other questions.**

  1. The On-AVVP subtask requires identifying audio-only, visual-only, and audio-visual events. Does the proposed method predict each type of event separately or simultaneously?
  2. Could the softmax function be used to normalize the weight w(t) across frames in Eq. 9?
  3. What are the training and inference time costs for the proposed method?

- **Minor issues (do not affect my rating):**

  1. The module name “current enhancement” in Line 157 is inconsistent with “current refinement” shown in Figure 2.

  2. In Figure 2, the "Focal Temporal Prioritization" module appears disconnected from the rest of the architecture. It would be clearer to illustrate how this module interacts with others.

  3. It would be helpful to include a diagram (even in the appendix) that explains the training and inference processes described in Section 3.6.

  4. The paper would benefit from citing more traditional works on audiovisual event localization and video parsing, such as: Contrastive Positive Sample Propagation Along the Audio-Visual Event Line, TPAMI 2023. UWAV: Uncertainty-weighted Weakly-supervised Audio-Visual Video Parsing, CVPR 2025



####

---

> ### Author Rebuttal · Authors · 2025-07-31
>
> Thank you for recognizing our work and for your constructive comment. In regard to the weaknesses and questions raised, we provide our point-to-point responses below:
>
> >**W1**:
> Contradictory statements? Lines 256–257 suggest that the PF module significantly boosts performance, whereas Lines 268–269 state that PF yields negligible gains. It would be helpful to clarify these descriptions to avoid confusion.
>
> **A1**:
>
> Thank you for pointing out the potentially confusing phrasing in Lines 268–269.
>
> In Lines 268–269, we wish to clarify that the performance benefits derived from our pseudo-future mechanism are attributable to the effective learning guided by these targeted future-oriented losses, rather than merely an increase in model capacity. The phrase “negligible gains” refers to the variant in which we **merely add the  $PF$  architecture’s parameters without applying any future supervision** (Row 2 in Tab.3(b) of our manuscript).
>
> We will revise the manuscript to express these points more clearly.
>
> ---
>
> >**W2**:
> Failure cases. While Figure 5 and the supplementary materials present visualizations, the paper would benefit from a more detailed discussion of the failure cases—when and why PreFM might not perform well.
>
>
> **A2**:
>
> Thank you for this excellent suggestion.
>
> We conduct a more detailed analysis on the On-AVEL test set to identify specific failure cases. While the rebuttal format prevents us from including new figures and PDF, we present our quantitative findings and analysis below:
>
> 1. **Confusion Between Similar Events**: We analyze the events with the lowest performance and their most common confusions.
>
> | Event | Precision | Recall | F1 | Most confused with |
> | -------- | -------- | -------- |-|-|
> |people slurping|0.55|0.17|0.25|people eating, man speaking|
> |people shouting|0.42|0.19|0.27|baby laughter, engine knocking|
>
> This results reveal that PreFM struggles to distinguish between events that are semantically or acoustically similar. We hypothesize this is because our current framework does not explicitly incorporate a contrastive learning design to better separate the representations of events originating from similar audio-visual sources.
>
> 2. **Performance in Dense Scenes**: We analyze the impact of event density (the number of event classes within a video) on event-level performance.
>
> | num_events | Avg F1 |
> | -------- | -------- |
> | 1-3     | 0.54     |
> |4-6|0.23|
> |>6|0.13|
>
> These results show that PreFM's performance degrade in complex videos containing a large number of distinct event classes. This suggests that while our future modeling is effective, its benefits are less pronounced in scenarios with very rapid scene changes and drastic context shifts.
>
> We will add this analysis and qualitative visualizations of these failure cases to the appendix of our revised manuscript.
>
> ---
> >**W3.1**:
> The On-AVVP subtask requires identifying audio-only, visual-only, and audio-visual events. Does the proposed method predict each type of event separately or simultaneously?
>
> **A3.1**:
>
> **Our method predicts all event types simultaneously.**
>
> We follow the standard paradigm for Audio-Visual Video Parsing (AVVP) tasks by treating it as a multi-label classification problem over a unified class set. Specifically, our model's final classifier produces a prediction vector of size $C_a+C_v$ to predict audio-only, visual-only and audio-visual event (an event is considered "audio-visual" when its auditory and visual components are both present) results, where $C_a$ and $C_v$ represent the number of audio and visual event classes, respectively.
>
>
>
> ---
> >**W3.2**:
> Could the softmax function be used to normalize the weight w(t) across frames in Eq. 9?
>
> **A3.2**:
>
> The softmax function can be used to normalize the weight $w(t)$, but we do not recommend it for our specific purpose. Our reasoning is as follows:
>
> * Our current normalization approach, after obtaining the Gaussian distribution weights, can be expressed as: $w(t) = \frac{g(t)}{\sum_{i} g(t_i)}$. This direct normalization **preserves the proportional relationship of the Gaussian distribution**, creating a smooth focus.
> * In contrast, the softmax function is expressed as: $w(t) = \frac{e^{g(t)}}{\sum_{i} e^{g(t_i)}}$. The exponential mapping in softmax would exaggerate the difference between high and low weight values. This would cause the model to **overly ignore the edge information** (i.e., the earliest historical context and the most distant future cues), which is not what we intend.
>
> Furthermore, we conduct an ablation study on the On-AVEL task where we replace our normalization method with softmax:
>
> |  | mAP | Avg F1|
> | -------- | -------- | -------- |
> | Ours     | 70.1     | 46.3     |
> | Ours w/ softmax | 70.0| 45.9 |
>
> As the results show, model performance slightly decreases after switching to softmax. This empirically supports our choice of using direct normalization.
>
>
> ---
> >**W3.3**:
> What are the training and inference time costs for the proposed method?
>
> **A3.3**:
>
> The total training time required to converge, inference processing time costs (latency) and model performance of our method are shown below. We use the official code to reproduce the second-best baseline methods CCNet on the On-AVEL and MM-CSE on the On-AVVP tasks to provide further context:
>
> |On-AVEL|Epoch|Training time|Latency↓|Avg F1↑|
> |-|-|-|-|-|
> |CCNet|45|11h 4mins|133.3ms|37.0|
> |PreFM|60|1h 1min|19.3ms|46.3|
> |PreFM+|60|1h 25mins|23.8ms|51.5|
>
> |On-AVVP|Epoch|Training time|Latency↓|Avg $_{av}$ F1↑|
> |-|-|-|-|-|
> |MM-CSE|60|53mins|27.7ms|36.2|
> |PreFM|60|45mins|10.6ms|41.2|
> |PreFM+|60|48mins|18.7ms|42.2|
>
> As the results demonstrate, our PreFM framework offers substantial improvements in training and inference time costs.
>
> ---
> >**W4**:
> Four minor issues.
>
> **A4**:
>
> Thank you very much for your careful reading and these valuable suggestions to improve our paper's clarity and completeness. We will address all of them in the revised manuscript:
>
> * **Module Name Inconsistency**: We will revise the module name in Line 157 from "current enhancement" to "current refinement" to ensure it is consistent with Fig.2.
> * **Clarity of Fig.2**: We will revise Fig.2 to visually connect the "Focal Temporal Prioritization" module to the final loss computation. This will better illustrate its role in weighting the training objective.
> * **Training/Inference Visualizations**: As suggested, we will add a new figure to the appendix that visualizes the training and inference processes as described in Sec 3.6.
> * **Additional Citations**: Thank you for pointing out these highly relevant papers. We will add and discuss the suggested works (CPSP and UWAV), along with other recent related papers, in our Related Work section to provide a more comprehensive background.
>
> We appreciate this detailed feedback, which will certainly help us improve the quality of our paper.

---

> > ### Comment · Reviewer_No1a · 2025-08-02
> >
> > I appreciate the authors' efforts during the rebuttal, which have satisfactorily addressed my questions. The newly added discussions on failure cases and training/inference efficiency further enhance the completeness and clarity of the paper. I have no remaining concerns for discussion in this period. I hope these improvements, including the authors’ responses to other knowledgeable reviewers, can be incorporated into the final version.
> >
> > Overall, this paper introduces a novel and timely task, proposes an effective and efficient baseline, and presents extensive experiments with thorough analysis. I consider this a strong submission and recommend acceptance. I maintain my original positive rating.

---

> > > ### Author Response · Authors · 2025-08-05
> > > **Response to Reviewer No1a**
> > >
> > > Thank you for your positive comments and recommendation to our work. We will make all essential improvements based on your and other reviewers' feedbacks.
> > >
> > > Best wishes!

---

### Official Review · Reviewer_VEsL · 2025-07-03

**Clarity:** 2
**Significance:** 3
**Originality:** 3
**Rating:** 5
**Confidence:** 4

**Summary:**

This paper introduces Online Audio-Visual Event Parsing (On-AVEP), a novel paradigm for real-time parsing of audio, visual, and audio-visual events in streaming video content. This paper proposes the Predictive Future Modeling (PreFM) framework that addresses two key challenges: accurate online inference with limited context and real-time computational efficiency.

**Questions:**

- Prediction Mechanism: Can you provide more technical details about how future audio-visual cues are predicted? What is the prediction horizon and how do you handle prediction uncertainty?
- Baseline Fairness: How were the baseline methods adapted for online processing? Are you comparing against methods specifically designed for online settings or offline methods retrofitted for streaming?
- Real-time Performance: Can you provide the concrete real-time performance? What are the actual processing times and computational costs compared to baseline methods in practical streaming scenarios?

**Ethical Concerns:**

["NO or VERY MINOR ethics concerns only"]

**Final Justification:**

I sincerely thank the authors for their patient response and the effort put into the rebuttal. My concerns have been addressed. and I will increase my score. I hope the authors can include the additional experimental results and analysis mentioned above in the paper to further improve its quality.

**Limitations:**

The paper lacks sufficient concrete analysis of its primary claimed advantages, particularly regarding the technical details of predictive modeling, component-wise contributions, and quantitative real-time performance validation.

**Paper Formatting Concerns:**

I sincerely thank the authors for their patient response and the effort put into the rebuttal. My concerns have been addressed, and I have increased my score. I hope the authors can include the additional experimental results and analysis mentioned above in the final version to further improve its quality.

**Quality:**

3

**Strengths And Weaknesses:**

Strength

This paper addresses an important practical problem in real-time multimodal processing. The proposed predictive future modeling approach is novel and the idea of anticipating future multimodal cues to compensate for limited historical context in streaming scenarios is technically sound. The claimed performance gains with fewer parameters suggest good computational efficiency.

Weaknesses

- About predictive multimodal future modeling: This paper lacks sufficient detail about the specific mechanisms of predictive future modeling. How exactly does the model predict future cues? What is the temporal horizon?
- Real-time Claims: The paper emphasizes real-time efficiency but doesn't provide concrete timing analysis, latency measurements. Claims about "real-time applicability" need quantitative support.
- Missing Ablation Studies: There's insufficient analysis of which components contribute most to the performance gains. The interplay between predictive multimodal future modeling, modality-agnostic representation, and focal temporal prioritization needs better investigation.

---

> ### Author Rebuttal · Authors · 2025-07-31
>
> Thank you for your positive feedback and valuable suggestions. In regard to the weaknesses and questions raised, we provide our point-to-point responses below:
>
> ---
> >**W1 and Q1**:
> Prediction Mechanism: How exactly does the model predict future cues? What is the temporal horizon? Can you provide more technical details about how future audio-visual cues are predicted? What is the prediction horizon and how do you handle prediction uncertainty?
>
> **A1**:
>
> Here we provide a clearer explanation of the predictive future modeling mechanism, temporal horizon and uncertainty handling:
>
> **1. Prediction Mechanism:**
>
> Initial Future Modeling: We first use a set of learnable query tokens ($Q_a , Q_v$) for future time steps. These queries attend to the current audio-visual features via an attention mechanism to generate an initial, context-aware pseudo-future ($\tilde{F}_f^a, \tilde{F}_f^v$ in Eq.3).
>
> Refinement via TMCF: This initial, and potentially noisy, pseudo-future is then refined within our Temporal-Modality Cross Fusion (TMCF) stage. Using unified hybrid attention (UHA) block, this step produces an augmented and more reliable future representation ($\hat{F}^a_f,\hat{F}^v_f$ in Eq.4 and Fig.3).
>
> Supervision: The entire model is learned end-to-end, supervised by losses on these future predictions, specifically the classification loss $\mathcal{L}\_f$ and the modality-agnostic representation loss $\mathcal{L}_{mrr}$. This process effectively teaches the model to map the current context to reliable future events.
>
>
> **2. Temporal Horizon**
>
> The prediction horizon (the length of the pseudo-future window) is a hyperparameter $L_f$ and our default setting is $L_f=5$ seconds. This choice is empirically validated by our ablation study in Tab.4 (in the appendix), which shows that $L_f=5$ achieves the best balance between providing sufficient future context and avoiding excessive predictive noise.
>
> **3. Handling Prediction Uncertainty**
>
> Our framework handles this inherent uncertainty through the **Temporal-Modality Cross Fusion (TMCF)** stage. TMCF refines the initial, potentially noisy pseudo-future by enforcing further cross-modal and cross-temporal interactions (as described in Lines 151-156 and Fig.3 in our manuscript), which can be broken down as follows:
> * Self-Attention: Models the internal temporal structure and consistency within the predicted future sequence.
> * Cross-Modal Attention: Ensures the predicted audio and visual future streams are coherent with each other.
> * Cross-Temporal Attention: Grounds the uncertain future prediction in the more reliable current context, effectively calibrating and denoising it.
>
> The effectiveness of this design is quantitatively demonstrated in Tab.3(c). The full TMCF achieves higher F1-scores on future predictions (e.g., **57.5** at T+1) compared to ablated versions (e.g., 55.4 with "Self" attention only), proving that our fusion mechanism successfully handles the uncertainty and improves the reliability of the pseudo-future context.
>
> ---
>
> >**W2 and Q3**:
> Claims about "real-time applicability" need quantitative support.: Can you provide the concrete real-time performance? What are the actual processing times and computational costs compared to baseline methods in practical streaming scenarios?
>
> **A2**:
>
> We conduct a detailed timing analysis of processing latency to further substantiate our real-time claims. For the reviewer's convenience, we present a comprehensive comparison below, including the performance, latency and other efficiency metrics in Tab.1 and Tab.2. All tests are conducted under a single RTX 3090 GPU.
>
> |On-AVEL|Avg F1↑|Params↓|FLOPs↓|Peak Memory↓|FPS↑|Latency↓|
> |-|-|-|-|-|-|-|
> |UnAV|28.6|139.4M|52.4G|764.7MB|10.6|94.3ms|
> |UniAV|30.3|130.8M|22.7G|1020.5MB|15.6|64.1ms|
> |CCNet|37.0|238.8M|72.1G|1179.4MB|7.5|133.3ms|
> |PreFM (Ours)|46.3|**6.5M**|**0.4G**|**56.4MB**|**51.9**|**19.3ms**|
> |PreFM+ (Ours)|**51.5**|13.8M|0.5G|144.2MB|42.0|23.8ms|
>
> |On-AVVP|Avg $_{av}$ F1 ↑|Params↓|FLOPs↓|Peak Memory↓|FPS↑|Latency↓|
> |-|-|-|-|-|-|-|
> |VALOR|33.0|4.9M|0.45G|**20.1MB**|62.2|16.1ms|
> |CoLeaF|29.7|5.7M|0.25G|114.1MB|60.4|16.6ms|
> |LEAP|34.3|52.0M|1.09G|204.7MB|19.3|51.8ms|
> |NERP|33.5|9.6M|1.69G|90.2MB|26.4|37.9ms|
> |MM-CSE|36.2|6.2M|0.91G|33.0MB|36.1|27.7ms|
> |PreFM (Ours)|41.2|**3.3M**|**0.22G**|20.7MB|**94.4**|**10.6ms**|
> |PreFM+ (Ours)|**42.2**|12.1M|0.48G|55.9MB|53.5|18.7ms|
>
> Taking the On-AVEL task as a representative example, we observe that:
>
> * **Low Latency**: PreFM has a latency of only **19.3 ms** per inference. This is approximately 7 times faster than the next best performing method, CCNet, which has a latency of 133.3 ms.
> * **Superior Efficiency**: This real-time performance is achieved while using fewer resources: **+9.3** Avg F1-score with only **2.7%** of the parameters and **0.6%** of the FLOPs compared to CCNet.
>
> In the revised manuscript, we will add these latency results and a concise discussion to fully substantiate our “real-time applicability” claims.
>
> ---
>
> >**W3**:
> Missing Ablation Studies: There's insufficient analysis of which components contribute most to the performance gains. The interplay between predictive multimodal future modeling, modality-agnostic representation, and focal temporal prioritization needs better investigation.
>
> **A3**:
>
> **1. Individual Component Contributions**
>
> To better isolate which component contributes most, we conduct additional ablation studies by adding each component individually to our baseline model (which uses a context window of $L_c$ but no other improvements). The results are summarized below:
>
> |Components|mAP|Avg F1|
> |-|-|-|
> |Baseline|69.1|40.8 (+0.0)|
> |+Pseudo Future Mechanism ($PF$)|69.7|42.4 (+1.6)|
> |+Focal Temporal Prioritization ($w(t)$)|69.3|41.1 (+0.3)|
> |+Modality-agnostic Robust Representation ($\mathcal{L}_{mrr}$)|69.2|41.7 (+0.9)|
> |+Random Sampling ($RS$)|69.8|41.8 (+1.0)|
> |+All|70.1|46.3 (+5.5)|
>
> This analysis clearly shows that **Pseudo Future Mechanism ($PF$) provides the most significant performance gain** when applied individually, underscoring its central role in our framework. Other components like $w(t)$ and $\mathcal{L}_{mrr}$ also provide substantial improvements.
>
> **2. The Interplay Among Components**
>
> Our original ablation study in Tab.3(a) is designed to reveal the interplay between components. Based on that table and the new results above, we provide a more focused analysis of their synergy:
>
> * **Synergy between $PF$ and $\mathcal{L}\_{mrr}$**: $PF$ alone improves the Avg F1-score by +1.6 and $\mathcal{L}\_{mrr}$ alone improves by +0.9. When combined $PF$ with $\mathcal{L}\_{mrr}$, it yields an improvement of +3.4 (row 5 in Tab.3(a) ). This strong synergistic effect suggests that while the $PF$ mechanism is the primary driver for performance by providing future context, the $\mathcal{L}_{mrr}$ acts as a powerful regularizer and guide. It enriches the pseudo-future predictions by forcing them to align with the robust, semantically-rich feature space of the teacher model, leading to a more effective and generalizable representation than either component could achieve alone.
> * **$w(t)$ as a Consistent Enhancer**: Adding $w(t)$ to the $PF$ model (row 4 vs. row 3 in Tab.3(a) ) boosts the gain from +1.6 to +2.2. Similarly, adding it to a model with $PF$ and $\mathcal{L}_{mrr}$ (row 7 vs. row 5 in Tab.3(a) ) elevates the gain from +3.4 to +4.6. This consistent improvement highlights the unique value of $w(t)$: by applying a Gaussian-weighted focus to the loss, it compels the model to prioritize making a precise and robust prediction at the most critical moment—the present $T$—thereby sharpening the final online decision-making process.
>
> ---
> >**Q2**:
> Baseline Fairness: How were the baseline methods adapted for online processing? Are you comparing against methods specifically designed for online settings or offline methods retrofitted for streaming?
>
> **A2**:
>
> **1. Justification: Establishing Baselines in a Clear Research Gap**
>
> We compare PreFM against SOTA offline methods that we adapt for the online setting. This is a deliberate and necessary choice due to a clear research gap in the field because:
> * Existing Audio-Visual Event Parsing methods (like CCNet for On-AVEL, or MM-CSE for On-AVVP) are exclusively offline and designed to process entire videos at once, possessing no native online capabilities.
> * Existing Online Video Understanding methods, conversely, are almost entirely unimodal (vision-only). They focus on tasks like action detection and are not equipped to handle the multimodal audio-visual parsing required by On-AVEP.
>
> Given this landscape, the most rigorous and scientifically valid approach is to adapt the strongest SOTA methods from the relevant offline domain to create the first powerful set of baselines for this new online task.
>
> **2. Fair Adaptation Protocol for Online Processing**
>
> We detail the fair adaptation protocol in the appendix (Lines 637-647) in our manuscript. For convenience, we summarize the process here:
>
> * For On-AVEL baselines, which require full-video inputs, we feed the model the available stream up to the current time $T$ and mask all future frames beyond $T$ with zero-padding.
> * For On-AVVP baselines, which operate on fixed-length clips, we feed the model the most recent segment ending at $T$ (e.g., from $T-9$ to $T$).
>
> This protocol guarantees that all methods, including our PreFM, **operate under the exact same online constraints, creating a fair and valid comparison.**

---

> > ### Comment · Reviewer_VEsL · 2025-08-06
> >
> > Thank you for the authors' efforts in addressing the rebuttal, which has resolved most of my concerns. Here are my further questions:
> >
> > 1) How is the parameter count of PreFM calculated? Why is it less than 1/20 of the parameters compared to UnAV/UniAV, yet the model still uses a significant number of self-attention layers and complex operations?
> >
> > 2) Does the calculation of FLOPs, latency, etc., include the time for feature extraction? After accounting for the feature extraction time, is real-time processing achievable?
> >
> > 3) There are now many powerful multi-modal large language models that can also understand audio, such as Qwen-omni, Gemini, and GPT-4o. How does the performance of this model compare to them?

---

> > > ### Author Response · Authors · 2025-08-07
> > > **Further Response to Reviewer VEsL**
> > >
> > > >**Q1:** How is the parameter count of PreFM calculated? Why is it less than 1/20 of the parameters compared to UnAV/UniAV, yet the model still uses a significant number of self-attention layers and complex operations?
> > >
> > > **A1**:
> > > 1. Parameter Count Calculation
> > >
> > > We calculate the trainable parameters for all models using a consistent protocol: **`sum(p.numel() for p in model.parameters() if p.requires_grad)`**, ensuring a fair comparison.
> > >
> > > 2. Lower Parameter Count
> > >
> > > The significant reduction in parameter count is a direct result of **PreFM's fundamentally different architecture** compared to UnAV, UniAV, and CCNet.
> > > * **Prior Methods with Large Transformer Backbone**: These models rely on large transformer backbones to process the whole sequences of video features and capture long-range temporal dependencies, requiring substantial parameters for full-context modeling.
> > > * **PreFM with lightweight UHA Block**: PreFM is built on a lightweight Universal Hybrid Attention (UHA) block (Eq.1 in our manuscript) designed for processing of short, fixed-length history windows, reducing parameter requirements while maintaining effectiveness.
> > >
> > > In summary, PreFM **does not use a significant number of attention layers**. Instead, it leverages only several UHA blocks that model future cues through fixed historical information, achieving superior performance with fewer parameters.
> > >
> > > ---
> > > >**Q2:** Does the calculation include the time for feature extraction? After accounting for the feature extraction time, is real-time processing achievable?
> > >
> > > **A2**:
> > > 1. Clarification on Efficiency Metrics
> > >
> > > Our metric calculations focus on **the core event parsing module**, excluding the feature extraction stage, for two key reasons:
> > > * **Benchmark Standard**: The metric calculations are based on the standard evaluation protocols of UnAV-100 and LLP, which only provide feature inputs for testing, making feature extraction external to the core parsing task evaluation.
> > > * **Fair Comparison**: All baseline methods are evaluated using the same protocol—taking pre-extracted features as input to test parsing capabilities. This ensures a fair comparison across different approaches.
> > >
> > > 2. Real-Time Processing with Feature Extraction
> > >
> > > **The real-time processing performance is achievable with PreFM:**
> > >
> > > We perform end-to-end tests on raw video (720p) and audio (48kHz) streams using CLIP and CLAP as feature extractors. The pseudo-code of the asynchronous pipeline is as follows:
> > > ```
> > > # Initialize CUDA streams and events
> > > v_stream, a_stream, prefm_stream = torch.cuda.Stream()
> > > v_event, a_event = torch.cuda.Event()
> > >
> > > time.start()
> > >
> > > with torch.no_grad():
> > >     # Process video stream and get image features
> > >     with torch.cuda.stream(v_stream):
> > >         v_tensor = processor(raw_image)
> > >         v_feat = clip_model.get_image_features(v_tensor)
> > >         v_event.record(v_stream)
> > >
> > >     # Process audio stream and get audio features
> > >     with torch.cuda.stream(a_stream):
> > >         a_tensor = processor(raw_audio)
> > >         a_feat = clap_model.get_audio_features(a_tensor)
> > >         a_event.record(a_stream)
> > >
> > >     # Process results
> > >     with torch.cuda.stream(prefm_stream):
> > >         prefm_stream.wait_event(v_event)
> > >         prefm_stream.wait_event(a_event)
> > >         results = prefm_model(v_feat, a_feat)
> > >
> > > time.end()
> > > ```
> > > This pipeline is tested on two distinct GPUs, and we present the end-to-end performance below, along with PreFM’s latency.
> > >
> > > |GPU|PreFM Latency|End-to-end Latency|End-to-end FPS|
> > > |-|-|-|-|
> > > |4060|19.5ms|40.7ms|24.6|
> > > |3090|19.3ms|36.5ms|27.4|
> > >
> > > These results clearly show that:
> > > * **Practical Real-Time Performance Achieved**: Our PreFM achieves **over 24 FPS** on both tested GPUs, meeting the conventional real-time processing standard.
> > > * **Scalable Performance for Applications**: Even on a consumer-grade GPU like the 4060, PreFM shows low latency, demonstrating broad applicability for various deployment scenarios.
> > > ---
> > > >**Q3:** How does the performance of PreFM compare to MLLMs models?
> > >
> > > **A3**:
> > >
> > > Thank you for your insightful suggestion.
> > >
> > > We respectfully note that direct **comparison would not be appropriate** between our work and MLLMs due to two main reasons:
> > > * **Different Focus and Task Scope**: Our PreFM focuses on online audio-visual event parsing in streaming video. In contrast, MLLMs are general models designed for broader tasks like multi-modal reasoning over large-scale datasets. The distinct nature of these tasks makes comparison **outside the scope of our current work**.
> > > * **Different Model Scale**: PreFM is a lightweight model (~6.5M parameters) designed for resource-constrained real-time scenarios. While MLLMs operate at billion-parameter scales where latency and model size are less constrained, making comparison **unfair and less meaningful**.
> > >
> > > We acknowledge the potential of incorporating MLLMs into the online audio-visual event parsing task, and we sincerely appreciate you highlighting this avenue for future research.
> > >
> > > Thank you once again for your time and constructive feedback.

---

> > > > ### Comment · Reviewer_VEsL · 2025-08-08
> > > >
> > > > I sincerely thank the authors for their patient response and the effort put into the rebuttal. My concerns have been addressed, and I will increase my score. I hope the authors can include the additional experimental results and analysis mentioned above in the paper to further improve its quality.

---

> ### Author Response · Authors · 2025-08-05
> **Response to Reviewer VEsL**
>
> Dear Reviewer VEsL,
>
> We greatly appreciate your time and consideration of our submission. As the author-reviewer discussion period is nearing its end, we kindly ask if you could review our responses to your comments at your earliest convenience.
>
> This will allow us to address any further questions or concerns you may have before the discussion period ends. If our responses satisfactorily address your concerns, please let us know.
>
> Thank you very much for your time and effort!
>
> Sincerely,
>
> The Authors of Submission #2457

---

### Official Review · Reviewer_UN6v · 2025-07-05

**Clarity:** 3
**Significance:** 3
**Originality:** 1
**Rating:** 4
**Confidence:** 4

**Summary:**

This paper introduces Online Audio-Visual Event Parsing (On-AVEP), a new setting for multimodal video understanding where models must parse audio, visual, and audio-visual events in a strictly online manner. To address the unique challenges of this setting—limited future context and real-time constraints—the authors propose PreFM, a lightweight framework that enhances online inference by predicting useful future multimodal cues and incorporating them during current-step inference. PreFM also introduces a modality-agnostic representation mechanism and a temporal focus strategy to improve both robustness and efficiency. The method shows strong performance with significantly fewer parameters, suggesting its practical utility for real-time applications.

**Questions:**

Please refer to weaknesses

**Ethical Concerns:**

["NO or VERY MINOR ethics concerns only"]

**Final Justification:**

The authors have been able to clarify and respond to the questions I had and I am wlliam to increase my score. I also read through other reviewer comments and their rebuttals to make better judgement overall, and I agree it should be accepted.

**Limitations:**

Yes

**Quality:**

2

**Strengths And Weaknesses:**

### Strengths:
- The method is intuitive and simple to understand.
- The problem is well motivated, and empirical results support the claims.
- The method appears to work well in online settings, with low memory footprint.

### Weaknesses and questions:
- Despite the results, the methodical contribution is quite weak. At its core, the work simply teases out \(k\) representations using some "prototype" query vectors via attention, as a proxy for the supposed future.
- The paper uses a series of attention blocks to generate future representations. However, the **order of operations** and **inputs to the UHA blocks in TMCF** appear fairly arbitrary.
  Can the authors provide justification for this design? For instance:
  - What motivates the specific sequence of UHA operations?
  - Why were those particular inputs chosen?
  - What happens if you simply apply another round of attention (as in the "Pseudo future" section)?

  It's unclear whether TMCF meaningfully contributes, or if the observed gains come primarily from increased parameter count (due to FFL) and additional attention layers like those used in the "Pseudo future" section.

- The paper does not clearly articulate its contributions.
  Can the authors explicitly state the contributions and novelties in the introduction?

- While the method does seem effective in online settings with low memory usage, it's important to **contextualize these improvements**.
  What were the bottlenecks in prior methods, and how does this work address them differently?
  Additionally, can the authors clarify how their parameter count compares to prior art, and justify the *huge* differences?

- What are the implications of the method on training time ?

Open to increasing my score, provided my queries are addressed

---

> ### Author Rebuttal · Authors · 2025-07-31
>
> Thank you for your positive feedback and constructive comments. In regard to the weaknesses and questions raised, we provide our point-to-point responses below:
>
> ---
>
> >**W1**:
> Despite the results, the methodical contribution is quite weak. At its core, the work simply teases out (k) representations using some "prototype" query vectors via attention, as a proxy for the supposed future.
>
> **A1**:
>
> Our core methodological contribution lies in proposing a novel framework, **PreFM**, which addresses the new and challenging task of **Online Audio-Visual Event Parsing (On-AVEP)** through the Predictive Future Modeling mechanism, not in using query vectors for future prediction.
>
> The Predictive Future Modeling mechanism **is not simply about "teasing out representations**": Learnable queries are trained to actively anticipate plausible future cues based on the current context. This is followed by cross-modal and cross-temporal interactions to augment the initial, potentially noisy pseudo-future, which is then integrated back to enhance the current representations.
>
> This simple yet effective framework achieves **SOTA performance with high efficiency**: It delivers a **+9.3** Avg F1-score improvement while using only **2.7%** of the parameters of the SOTA model.
>
> ---
> >**W2**:
> What motivates the specific sequence of UHA operations? Why were those particular inputs chosen? What happens if you simply apply another round of attention (as in the "Pseudo future" section)?
>
>
> **A2**:
>
> **1. Justification for the TMCF Design**
>
> TMCF design is not arbitrary but is motivated by the core challenge of refining the noisy initial pseudo-future sequences. The key design principles are:
>
> **Parallel Operation in UHA**: The UHA block processes multiple context sets in parallel (by summing their attention outputs as Eq.1 in our manuscript), not in a rigid sequence.
>
> Logical Flow in TMCF: The whole TMCF process involves (a) Future Augmentation first, followed by (b) Current Refinement. This order is crucial: we must first refine the initial, potentially noisy pseudo-future to make them more reliable. Only then can this higher-quality future context be effectively integrated back to enhance the current representations.
>
> **Principled Input Selection**: The inputs for the UHA block in TMCF are chosen to enable three critical interactions for robust future modeling (as described in Lines 151-156 and Fig.3 in our manuscript):
>
> * Self-Attention: To model the internal temporal structure within the predicted future sequence.
> * Cross-Modal Attention: To ensure the predicted audio and visual futures are coherent with each other.
> * Cross-Temporal Attention: To ground the uncertain future prediction in the reliable, immediate past, thereby calibrating and denoising it.
>
> This multi-faceted fusion is essential for generating a high-quality, reliable future context.
>
> **2. Contribution of TMCF vs. Another Round of Attention**
>
> To empirically validate that the gains come from our specific design rather than merely an increased parameter count, we conduct a comprehensive ablation study, expanding upon Tab.3\(c). We compare our full TMCF against several variants: convert all modality fusion into temporal fusion (T only), convert all temporal fusion into modality fusion (M only) and convert all fusion layer into self-attention (Self), including the reviewer's suggestion of "simply applying another round of attention" (Simple Self).
>
> |Methods|Params|T+1|T+2|T+3|Avg F1|
> |-|-|-|-|-|-|
> |Simple Self|4.4M|54.5|53.6|52.8|44.1|
> |Self|6.5M|55.4|54.6|53.7|44.6(+0.0)|
> |M only|6.5M|56.7|56.0|55.5|45.3(+0.7)|
> |T only|6.5M|56.6|55.7|55.1|45.0(+0.4)|
> |TMCF|6.5M|**57.5**|**56.5**|**55.4**|**46.3(+1.7)**|
>
> The results show that simply adding a round of self-attention  performs poorly, and our full TMCF outperforms the baseline by a significant margin (+1.7 Avg F1) with the same parameter count. This strongly demonstrates that **the performance gain is attributable to the principled fusion mechanism of TMCF, not just the additional parameters.**
>
> ---
> >**W3**:
> The paper does not clearly articulate its contributions.
> Can the authors explicitly state the contributions and novelties in the introduction?
>
> **A3**:
>
> Our main contributions are three-fold:
>
> 1. We introduce **Online Audio-Visual Event Parsing (On-AVEP)**, a new paradigm for real-time multimodal understanding. To our knowledge, this is the first work to systematically address the challenge of parsing audio, visual, and audio-visual events from streaming video. We further establish that success in this paradigm requires two critical capabilities: (a) accurate online inference from limited context, and (b) real-time efficiency to balance performance with computational cost.
> 2. We propose the **PreFM** framework, a novel and efficient architecture for On-AVEP. PreFM's core innovations include: (a) **Predictive Multimodal Future Modeling** mechanism to overcome the critical problem of missing future context ; and (b) a combination of **Modality-agnostic Robust Representation** and **Focal Temporal Prioritization** to enhance model robustness and efficiency during training, providing an insightful approach to multimodal real-time video understanding.
> 3. We establish **new SOTA performance with unprecedented efficiency**. Extensive experiments on two public datasets show that PreFM drastically outperforms previous methods (e.g., **+9.3** Avg F1-score on UnAV-100), while using a fraction of the computational resources (e.g., only **2.7%** of the parameters of the next best model), validating it as a powerful and practical solution.
>
> We thank the reviewer for this constructive suggestion, and we will add the summary to the end of our introduction in the revised manuscript.
>
> ---
> >**W4**:
> What were the bottlenecks in prior methods, and how does this work address them differently?
> Can the authors clarify how their parameter count compares to prior art, and justify the huge differences?
>
> **A4**:
>
> **1. Bottlenecks in Prior Methods and Our Solutions**
>
> The bottlenecks of prior methods in an online setting are two-fold:
>
> * Fundamental Offline Design: These models are designed to process the entire video sequence to leverage global context for video understanding, which is fundamentally infeasible in a real-time streaming scenario.
> * Prohibitive Computational Cost: Their reliance on processing long sequences with heavy model architectures results in high latency and memory usage, failing to meet the demands of real-time applications.
>
> Our PreFM framework is designed to address these bottlenecks:
>
> * **Natively Online Architecture**: PreFM operates on a short, sliding window of past information, making it naturally suited for streaming data.
> * **Context Enrichment via Future Prediction**: Instead of requiring global context, PreFM's core innovation, Predictive Future Modeling, explicitly infers the necessary future cues from the local window. This enriches the limited context for robust real-time parsing.
> * **Training for Efficiency**: This is complemented by Modality-agnostic Robust Representation and Focal Temporal Prioritization, which boost inference accuracy while maintaining a lightweight training process.
>
> **2. Parameter Count and Justification**
>
> We calculate trainable parameters for all models using a consistent protocol (`sum(p.numel() for p in model.parameters() if p.requires_grad)`) for a fair and transparent comparison. We show the parameter results in Tab.1 and Tab.2 in our manuscript. Below is a summary for the reviewer's convenience:
>
> |On-AVEL|Params↓|
> |-|-|
> |UnAV|139.4M|
> |UniAV|130.8M|
> |CCNet|238.8M|
> |PreFM (Ours)|6.5M|
> |PreFM+ (Ours)|13.8M|
>
> |On-AVVP|Params↓|
> |-|-|
> |VALOR|4.9M|
> |CoLeaF|5.7M|
> |LEAP|52.0M|
> |NERP|9.6M|
> |MM-CSE|6.2M|
> |PreFM (Ours)|3.3M|
> |PreFM+ (Ours)|12.1M|
>
>
> The significant difference in parameter counts stems from: Prior Methods (e.g., UnAV, CCNet) **rely on a large transformer backbone** to process a very long sequence of video features and capture long-range dependencies across different time spans. PreFM, in contrast, **is built upon a custom-designed, lightweight UHA block**. The purpose of the UHA block is to efficiently perform two targeted tasks on a short history window: fusing current temporal-modal features and intelligently modeling the future.
>
>
> ---
> >**W5**:
> What are the implications of the method on training time ?
>
> **A5**:
>
> **PreFM offers a compelling advantage in training time.**
>
> We use the official code to reproduce the second-best baseline methods CCNet on the On-AVEL and MM-CSE on the On-AVVP tasks. These results, including the number of epochs, total training time required to converge, and performance, are summarized in the table below. All tests are conducted on a single RTX 3090 GPU:
>
> |On-AVEL|Epoch|Training time|Avg F1|
> |-|-|-|-|
> |CCNet|45|11h 4mins|37.0|
> |PreFM|60|1h 1min|46.3|
> |PreFM+|60|1h 25mins|51.5|
>
> |On-AVVP|Epoch|Training time|Avg $_{av}$ F1|
> |-|-|-|-|
> |MM-CSE|60|53mins|36.2|
> |PreFM|60|45mins|41.2|
> |PreFM+|60|48mins|42.2|
>
> Due to its lightweight and efficient design philosophy, Our PreFM framework demonstrates a significant advantage in training efficiency.

---

### Decision · Program_Chairs · 2025-09-17

**Decision:**

Accept (poster)

**Comment:**

This paper introduces a new task, Online Audio-Visual Event Parsing, which requires detecting audio, visual, and audio-visual events in videos under online constraints (i.e., using only past and present context). To address the lack of future information and the need for efficiency, the authors propose PreFM (Predictive Future Modeling).

Overall, this work defines a novel and important problem, proposes a solution that is both effective and efficient, and supports its claims with extensive experiments and analyses. While reviewers initially raised concerns about contribution clarity, baseline fairness, and practical evaluation, the authors provided convincing rebuttals with new results that addressed these issues. The combination of task novelty, methodological soundness, and strong empirical results makes this paper a solid contribution. All reviewers were satisfied after the rebuttal, and no unresolved concerns remain.  The AC concurs with the reviewers and recommends acceptance. The authors should incorporate the new results and discussions into the final version of the paper.